# LncEGFL7OS regulates human angiogenesis by interacting with MAX at the EGFL7/miR-126 locus

Qinbo Zhou[1†], Bo Yu[1†*], Chastain Anderson[1], Zhan-Peng Huang[2], Jakub Hanus[1], Wensheng Zhang[3], Yu Han[4], Partha S Bhattacharjee[5], Sathish Srinivasan[6], Kun Zhang[3], Da-zhi Wang[2], Shusheng Wang[1,7*]

[1]Department of Cell and Molecular Biology, Tulane University, New Orleans, United States; [2]Department of Cardiology, Boston Children's Hospital, Harvard Medical School, Boston, United States; [3]Department of Computer Science, Xavier University, New Orleans, United States; [4]Aab Cardiovascular Research Institute, University of Rochester School of Medicine and Dentistry, Rochester, United States; [5]Department of Biology, Xavier University, New Orleans, United States; [6]Cardiovascular Biology Research Program, Oklahoma Medical Research Foundation, Oklahoma, United States; [7]Department of Ophthalmology, Tulane University, New Orleans, United States

*For correspondence:
byu@tulane.edu (BY);
swang1@tulane.edu (SW)

[†]These authors contributed equally to this work

Competing interests: The authors declare that no competing interests exist.

**Abstract** In an effort to identify human endothelial cell (EC)-enriched lncRNAs,~500 lncRNAs were shown to be highly restricted in primary human ECs. Among them, *lncEGFL7OS*, located in the opposite strand of the *EGFL7/miR-126* gene, is regulated by ETS factors through a bidirectional promoter in ECs. It is enriched in highly vascularized human tissues, and upregulated in the hearts of dilated cardiomyopathy patients. LncEGFL7OS silencing impairs angiogenesis as shown by EC/fibroblast co-culture, in vitro/in vivo and ex vivo human choroid sprouting angiogenesis assays, while lncEGFL7OS overexpression has the opposite function. Mechanistically, lncEGFL7OS is required for MAPK and AKT pathway activation by regulating EGFL7/miR-126 expression. MAX protein was identified as a lncEGFL7OS-interacting protein that functions to regulate histone acetylation in the EGFL7/miR-126 promoter/enhancer. CRISPR-mediated targeting of EGLF7/miR-126/lncEGFL7OS locus inhibits angiogenesis, inciting therapeutic potential of targeting this locus. Our study establishes lncEGFL7OS as a human/primate-specific EC-restricted lncRNA critical for human angiogenesis.
DOI: https://doi.org/10.7554/eLife.40470.001

## Introduction

Angiogenesis plays a critical role in tissue development and homeostasis. Aberrant angiogenesis has been associated with numerous diseases, including heart disease, tumor growth, metastasis and age-related macular degeneration (AMD) (*Carmeliet, 2003*). Defective vascularization, usually associated with compensatory angiogenesis and vasculogenesis, has been observed in human dilated cardiomyopathy (DCM) patients (*Roura et al., 2007*; *Gavin et al., 1998*; *De Boer et al., 2003*). Methods to augment angiogenesis have been tested clinically for DCM (*Ylä-Herttuala et al., 2017*). Anti-angiogenic therapy, such as antibodies to vascular endothelial growth factors (VEGF), has shown efficacy clinically in treating wet AMD, the leading blinding disease in the elderly (*Brown et al., 2006*; *Rosenfeld et al., 2006*; *Zampros et al., 2012*; *Hurwitz et al., 2004*). However, some patients failed to respond to anti-VEGF treatment. Similarly, anti-angiogenic therapies have shown efficacy in certain cancers when used alone or combined with chemotherapy (*Miller et al.,*

**eLife digest** A well-networked supply of blood vessels is essential for delivering nutrients and oxygen to the body. To do so, new blood vessels need to form throughout life, from embryonic development to adult life. This process, known as angiogenesis, also plays a critical role in exercise, menstruation, injury and disease.

If it becomes faulty, it can lead to conditions such as the 'wet' version of age-related macular degeneration, where leaky blood vessels grow under the retina. This can lead to rapid and severe loss of vision. One way to treat this condition is to stop the growth of new blood vessels into this area using anti-angiogenic therapy, but not all patients respond to it. Identifying new mechanisms at play in human angiogenesis could provide insight into potential new therapies for this disease and other angiogenesis-related conditions.

A large amount of our genetic material is made up of a group of molecules called long non-coding RNAs or lncRNAs for short, which normally do not code for proteins. However, they are thought to play a role in many processes and diseases, but it has been unclear if they also influence angiogenesis. Now, Zhou, Yu et al. set out to study these RNA molecules in different types of human vessel-lining cells and to identify their role in angiogenesis.

Out of the 30,000 lncRNAs measured, about 500 of them were more abundant in these cells than other types of cells. One of the lncRNAs, called lncEGFL7OS, can be found on two human genes known to be relevant in angiogenesis (EGFL7 and miR-126). The results showed that patients with a condition called dilated cardiomyopathy, in which the heart muscle overstretches and becomes weak, had elevated levels of lncEGFL7OS. Other experiments analyzing human eye tissue revealed that lncEGFL7OS is required for angiogenesis by increasing the concentration of the EGFL7 and miR-126 gene products in cells. To achieve this, it binds together with a specific protein to the regulatory regions of the two genes to control their activity.

The discovery of a new control mechanism for angiogenesis in humans could lead to new therapies for conditions such as macular degeneration and other diseases in which angiogenesis is affected. A next step will be to see if the same RNA molecules and genes are also elevated in other diseases.

DOI: https://doi.org/10.7554/eLife.40470.002

2007; Sandler et al., 2006). However, anti-angiogenic therapy has met several hurdles on its way to be an main option for cancer therapy, mainly due to drug resistance (Shojaei and Ferrara, 2007). Identifying novel human angiogenesis mechanism would provide important insights and potential therapeutic options for angiogenesis-related diseases.

It is now established that up to 90% of the human genome is transcribed, and the majority of these transcripts are non-coding RNAs (ncRNAs) that do not encode proteins (Kapranov et al., 2007; Gerstein, 2012; Ecker, 2012). NcRNAs can be classified as short noncoding RNAs such as microRNAs (miRNAs), long noncoding RNAs (lncRNAs) and other classic ncRNAs. miRNAs include a group of small noncoding RNAs sized ~22 nucleotides that play important regulatory functions in numerous physiological and pathological processes, including angiogenesis (Wang and Olson, 2009). LncRNAs represent a large group of long (typically >200 nt) noncoding RNAs, whose function is still largely enigmatic (Ulitsky and Bartel, 2013). The study of lncRNAs in vascular biology is still in its infancy (Yu and Wang, 2018; MM and Goyal, 2016). Several lncRNAs, including MALAT1 (Liu et al., 2014), MANTIS (Leisegang et al., 2017), PUNISHER (Kurian et al., 2015), MEG3 (He et al., 2017; Qiu et al., 2016), MIAT (Yan et al., 2015), SENCR (Boulberdaa et al., 2016), GATA6-AS (Neumann et al., 2018) and STEEL (Man et al., 2018), have been shown to regulate angiogenesis. Dependent on their subcellular localizations, these lncRNAs function by regulating promoter and enhancer activities of angiogenesis-related genes in cis, or modulating gene expression by in trans mechanism through interaction with DNA/RNA-binding proteins or chromatin modifying proteins, or functioning as antisense RNAs to mRNAs or sponge for miRNAs in the cytoplasm.

By profiling more than 30,000 lncRNAs in several primary human EC lines, we have identified ~500 human EC-restricted lncRNAs. Among them, we focused on lncEGFL7OS, which is located in the opposite strand of the EGFL7/miR-126 gene. Through a series of in vitro and in vivo

experiments, we established lncEGFL7OS as a disease-relevant, human/primate-specific, EC-enriched lncRNA that is critical for angiogenesis through regulating MAX transcription factor activity at the EGFL7/miR-126 locus.

## Results

### Microarray profiling of lncRNAs in ECs and confirmation of the EC-restricted lncRNAs

To identify lncRNAs specific in ECs, a microarray was performed to profile ~30,000 lncRNAs and ~26,000 coding transcripts using an Arraystar human LncRNA microarray v3.0 system (Arraystar, Rockville, MD). Three primary human EC lines and two non-EC lines at low passages, namely, human umbilical vein EC (HUVEC), human retinal EC (HREC), human choroidal EC (HCEC), human dermal fibroblast cell (HDF) and human retinal pigment epithelial (RPE) cell lines, were used in the array. Purity of EC lines was confirmed by acetyl-LDL uptake and EC marker staining (*Figure 1—figure supplement 1*). Hierarchical cluster analysis of the array results validated the clustering of EC lines, which clearly separates from the HDF and RPE cell lines based on lncRNA and mRNA expression (*Figure 1A*). Moreover, lncRNAs appeared to be a stronger classifier to distinguish between EC and non-ECs than mRNAs. 498 lncRNAs are enriched in all three EC lines for more than two folds compared to the non-ECs (see *Figure 1B* for top 50 hits, *Supplementary file 1*). Among them, 308 are intergenic lncRNAs, 62 are sense overlapping lncRNAs, 83 are antisense lncRNAs, 23 are bidirectional lncRNAs, and 22 lncRNAs were previously identified as pseudogenes (*Figure 1C*). When these lncRNAs were cross-referenced with the enhancer-like lncRNAs, 19 of them are known enhancer-like lncRNAs with nearby coding genes within 300 kb (*Supplementary file 2*) (*Ørom et al., 2010*). We also took advantage of our microarray system in profiling both lncRNAs and mRNAs, and examined the lncRNA/mRNA regulation relationship for the EC-restricted lncRNAs. Since many lncRNAs have been shown to exert locus-specific effect on nearby genes, we first did a bioinformatics search for protein-coding genes that are within 10 kb of the 498 EC-restricted lncRNAs. 91 lncRNAs have protein-coding genes within 10 kb of the lncRNA gene (*Supplementary file 3*). Moreover, 27 of the 91 lncRNAs exhibited parallel expression pattern to the neighboring mRNAs in all 5 cell lines tested, while three of them showed inverse expression pattern relationship with the neighboring mRNAs. For some lncRNAs, including those near to *SRGN*, *FOXC2*, *STEAP1B*, *ECE1*, *GOT2*, *EGFL7* and *PRKAR1B*, the specificity for lncRNA in ECs is more robust than the neighboring mRNAs; for some other lncRNAs, including those near to *HHIP*, *ESAM*, and *UBE2L3*, their EC-specificity is less robust than their neighboring mRNAs. These results suggest that some lncRNAs can serve as robust EC-restricted gene expression markers. We also carried out a functional enrichment analysis based on the EC-restricted lncRNAs and the associated genes. The following biological processes and genes are highly represented in the associated lncRNAs with a false discovery rate (FDR) of less than 10% (*Figure 1—figure supplement 2A*): (1) heart development (NRP1, ECE1, FOXC2, PKD1, ZFPM2, FKBP1A, FOXP4); (2) chordate embryonic development (GATA2, SATB2, ECE1, LMX1B, FOXC2, PKD1, ZFPM2); (3) embryonic development ending in birth (GATA2, SATB2, ECE1, LMX1B, FOXC2, PKD1, ZFPM2); (4) blood vessel development (NRP1, EGFL7, ROBO4, FOXC2, PKD1, ZFPM2); (5) vasculature development (NRP1, EGFL7, ROBO4, FOXC2, PKD1, ZFPM2); and (6) metallopeptidase activity (ECE1, ADAMTS16, LTA4H, MMP25, ADAM15). From above, genes involved in embryonic development, especially vascular development, are associated with the EC-restricted lncRNAs. Taken together, we have established the lncRNA expression profile in ECs by comparative lncRNA microarray, and identified hundreds of EC-restricted lncRNAs, with a list of them having associated genes involved in vascular development.

Quantitative (q) RT-PCR was used to confirm a selected list of EC-enriched lncRNAs from the microarray. Friend leukemia integration 1 (FLI1) antisense lncRNA (FLI1AS, also named as SENCR (*Bell et al., 2014*), ASHGA5P026051), GATA binding protein 2 (GATA2) antisense lncRNA (lncGATA2, ASHGA5P019223, RP11-475N22.4), endothelial converting enzyme 1 (ECE1) intron sense-overlapping lncRNA (lncECE1, ASHGA5P032664, AX747766), endothelial cell-selective adhesion molecule (ESAM) bidirectional lncRNA (lncESAM, ASHGA5P021448, RP11-677M14.3), roundabout homolog 4 (ROBO4) nature antisense RNA (lncROBO4, ASHGA5P026882, RP11-664I21.5), and epidermal growth factor-like domain 7 (EGFL7) opposite strand lncRNA (lncEGFL7OS,

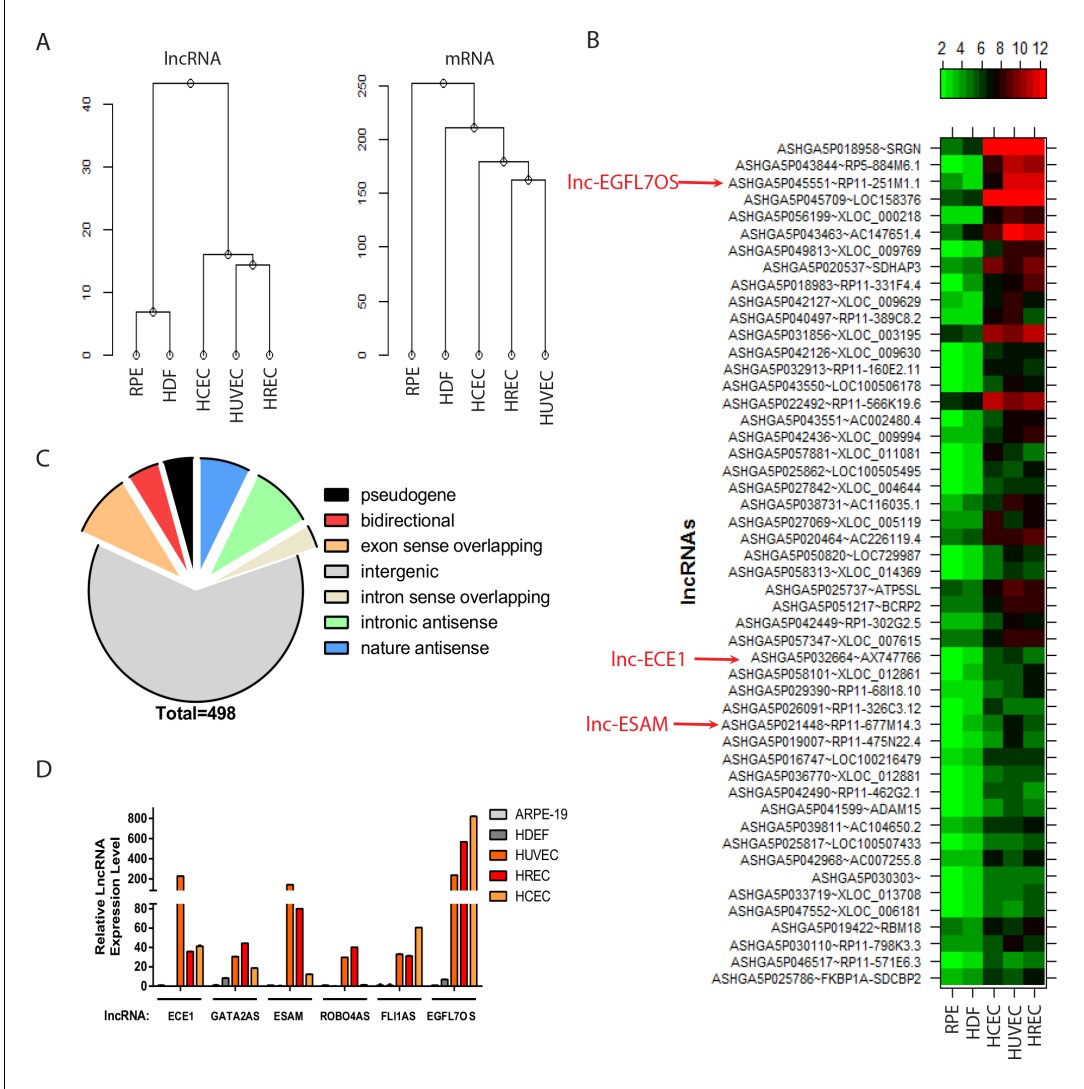

**Figure 1.** lncRNA profiling in ECs. (A) Hierarchy cluster analysis of lncRNA and mRNA expression data from five different cell lines. (B) Heatmap showing the top-50 enriched lncRNAs in three EC lines compared to the two non-EC lines. Several highlighted lncRNAs were used in the subsequent qRT-PCR confirmation in *Figure 1D*. (C) A pie chart showing different classes of annotated lncRNAs that are enriched more than two folds in ECs compared to non-ECs. (D) Quantitative (q) RT-PCR confirmation of candidate EC-enriched lncRNAs. n = 3. Error bars represent the standard error from three technical repeats from each line. GAPDH was used as normalization control.

DOI: https://doi.org/10.7554/eLife.40470.003

The following source data and figure supplements are available for figure 1:

**Source data 1.** *Figure 1D* source data.
DOI: https://doi.org/10.7554/eLife.40470.006
**Figure supplement 1.** EC Marker staining of the EC cells used.
DOI: https://doi.org/10.7554/eLife.40470.004
**Figure supplement 2.** Functional encrichment analysis and tissue distribution of the EC-enriched lncRNAs.
DOI: https://doi.org/10.7554/eLife.40470.005

ASHGA5P045551, RP11-251M1.1) were chosen because of their EC restriction and potential relevance to EC function. As shown in *Figure 1D*, the expression of lncECE1, lncGATA2, lncESAM, lncROBO4, lncFLI1 and lncEGFL7OS was found to be highly enriched in EC cell lines compared to the non-EC lines. Among different EC lines, lncECE1 and lncESAM were more enriched in HUVECs, while FLI1AS and lncEGFL7OS were more enriched in HCECs, supporting heterogeneity of ECs and suggesting differential expression of the lncRNAs in different ECs.

We also used a bioinformatics approach to determine the tissue distribution of the EC-restricted lncRNAs. The tissue expression information of the top 50 EC-restricted lncRNAs was obtained from the Stanford Source database (*Diehn et al., 2003*). *Figure 1—figure supplement 2B* shows the tissue distribution heatmap of the candidate lncRNAs with information available. The majority of the lncRNAs are enriched in the lung and placenta, which are highly vascularized tissues. Taken together, these data support the EC- and vasculature- restriction of the candidate lncRNAs from our microarray.

## Expression of lncEGFL7OS in human tissues and DCM patients

Given the involvement of EGFL7/miR-126 locus in regulating angiogenesis, we focused on lncEG-FL7OS, which partially overlaps with EGFL7/miR-126 gene but is transcribed in opposite direction (*Figure 2A*) (*Fish et al., 2008*; *Wang et al., 2008a*; *Kuhnert et al., 2008*; *Durrans and Stuhlmann, 2010*; *Parker et al., 2004*; *Schmidt et al., 2007*). The existence of lncEGFL7OS was confirmed by RT-PCR cloning using human placental RACE-ready cDNAs and subsequent sequencing, and the size of lncEGFL7OS is consistent with deposited gene AF161442 (*Figure 2—figure supplement 1A*). Interestingly, conserved homologous sequence of lncEGFL7OS only exists in humans and primates Rhesus monkey, but not in other lower vertebrate species including mice, suggesting lncEG-FL7OS is an evolutionarily new gene in mammals. We performed qRT-PCR to examine the tissue expression pattern of lncEGFL7OS. LncEGFL7OS was found to be highly enriched in the human lung, placenta and heart, which are highly vascularized tissues (*Figure 2B*). Since lncEGFL7OS overlaps with *EGFL7/miR-126*, the expression of *EGLF7* and miR-126 was also examined in parallel to lncEGFL7OS. Human *EGFL7* has four isoforms, named as *EGFL7A-D*, but only EGFL7B and EGFL7C are detectable by RT-PCR in human tissues. By qRT-PCR, both EGFL7B and EGFL7C are enriched in heart, kidney, bone marrow, uterus, thymus, thyroid, small intestine and placenta. Besides that, EGFL7B is more enriched in prostate, while EGFL7C is more enriched in lung and brain, suggesting a differential expression pattern of EGFL7 isoforms in humans (*Figure 2—figure supplement 1B*). miR-126 is highly enriched in the bone marrow, lung and heart (*Figure 2—figure supplement 1C*). Taken together, these results suggest there are both common and differential expression pattern of lncEGFL7OS and EGFL7/miR-126 in different human tissues.

We also examined the subcellular localization of lncEGFL7OS using both semi-quantitative RT-PCR and high-resolution RNA fluorescence in situ hybridization (FISH). By RT-PCR, lncEGFL7OS was shown to be expressed in both the cytoplasm and nucleus, but more in the nucleus of HUVECs (*Figure 2C*). SENCR was used a marker for cytoplasmic-enriched lncRNA, while NEAT-1 was used as a marker for nuclear enriched- lncRNA (*Bell et al., 2014*; *Zhang et al., 2013*). These results were confirmed by high-resolution RNA FISH experiment. RNA FISH with single-molecule sensitivity was performed using oligonucleotide (oligo) probe pools specific for lncEGFL7OS (*Cabili et al., 2015*). We observed variable numbers of lncEGFL7OS molecules in both the nucleus and cytoplasm of HUVECs (*Figure 2D*). RNaseA-treated samples were used as negative control and adeno-lncEG-FL7OS-overexpressed HUVECs were used as positive control for specificity of the probe. By quantification, the average copy number of lncEGFL7OS RNA in HUVECs is ~19, which is in agreement with the copy number (23–28 copies) by qRT-PCR using in vitro transcribed lncEGFL7OS as control for copy number calculation (*Supplementary file 4*). Taken together, these data indicate that lncEG-FL7OS is expressed at relatively low copy numbers in both the nucleus and cytoplasm of HUVEC cells.

To study the involvement of lncEGFL7OS in cardiovascular disease, we asked whether lncEG-FL7OS expression correlates with human dilated cardiomyopathy (DCM), a disease with defective vascularization (*Roura et al., 2007*; *Gavin et al., 1998*; *De Boer et al., 2003*). Increased expression of proangiogenic factors, including hypoxia-inducible factor 1α (HIF-1α) and VEGF-A, have been found in DCM, likely due to the compensatory angiogenesis and/or increased mobilization of endothelial progenitor cells (EPCs) to the diseased heart (*Roura et al., 2007*). The expression of lncEG-FL7OS was examined by qRT-PCR in the hearts of 7 DCM patients, with five healthy hearts used as controls. In the DCM hearts, the expression of atrial natriuretic peptide (ANP), a prominent marker for heart failure, was drastically upregulated (*Figure 2—figure supplement 1D*). In line with the increased angiogenic factors, the expression of EC/EPC marker PECAM-1 was also marginally increased. We found lncEGFL7OS expression was significantly elevated in the hearts of DCM

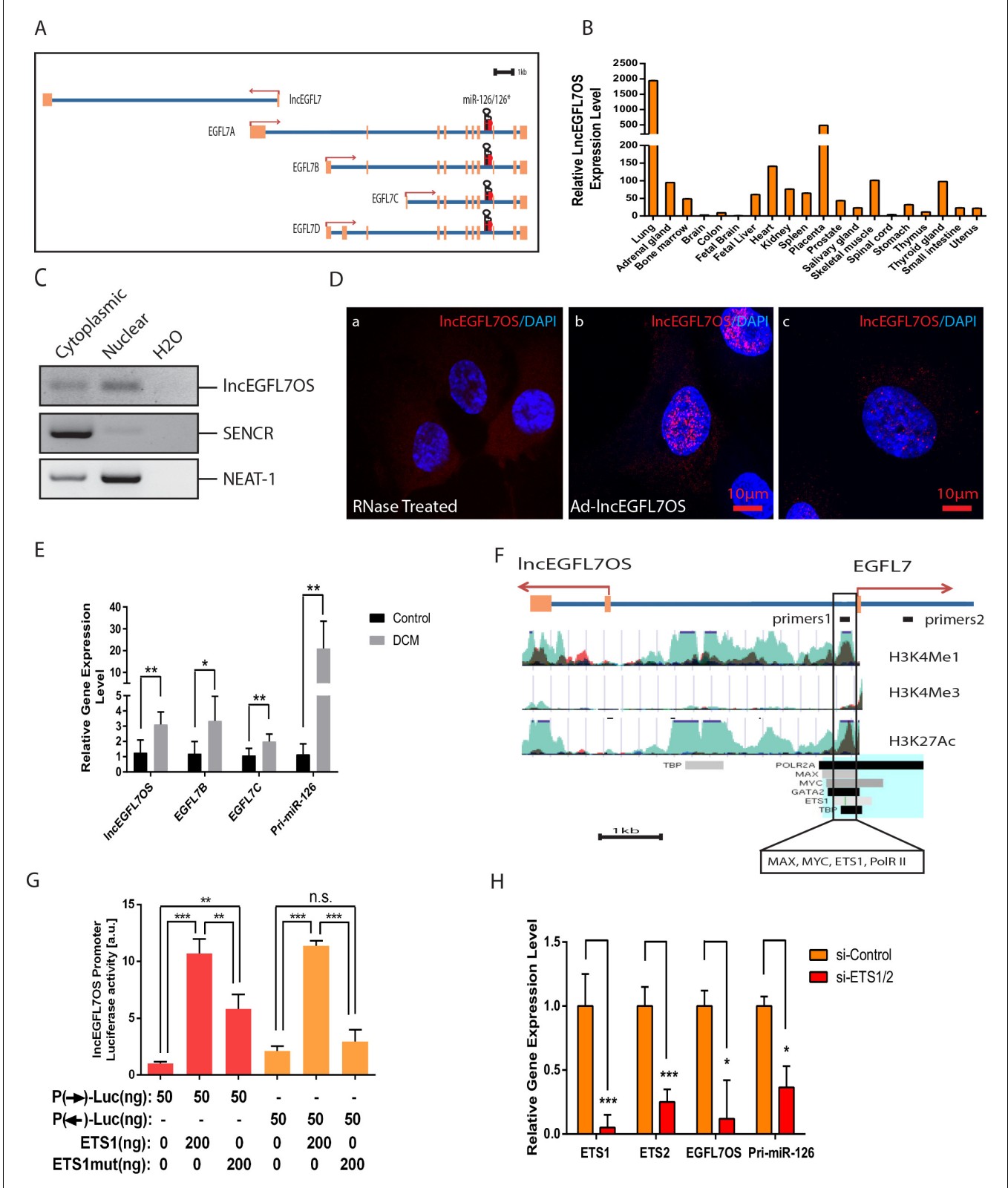

**Figure 2.** Expression, regulation and subcellular localization of lncEGFL7OS, as well as its regulation in DCM patients. (**A**) Genomic organization of lncEGFL7OS and its host gene EGFL7/miR-126. Exons are shown in orange and the introns are shown in blue. Direction of gene transcription is indicated by arrows. Scale = 1 kb; (**B**) Relative lncEGFL7OS expression level in different human tissues. GAPDH served as the normalization control. (**C**) Expression of lncEGFL7OS in the nucleus and cytoplasm of HUVECs shown by semi-quantitative RT-PCR. RT-PCR showing nuclear and cytoplasmic

*Figure 2 continued on next page*

*Figure 2 continued*

expression of lncEGFL7OS. SENCR was used a marker for cytoplasmic-enriched lncRNA, while NEAT-1 was used as a marker for nuclear-enriched lncRNA. (D) Expression of lncEGFL7OS in the nucleus and cytoplasm of HUVECs shown by high-resolution RNA FISH analysis (a–c). RNaseA-treated samples were used as negative control (a) and Ad-lncEGFL7OS-overexpressed HUVECs were used as positive control (b). Scale Bar equals 10 μm. (E) Upregulation of lncEGFL7OS, EGFL7B and C, and pri-miR-126 om the hearts of DCM patients. *p<0.05; **p<0.01. N = 6 for control samples and N = 7 for DCM samples. (F) Schematic potential promoter region (boxed) for EGFL7/lncEGFL7OS. Exons are shown in orange and the introns are shown in blue. Direction of gene transcription is indicated by red arrows. The peaks show regions with elevated H3K4Me1, H3K4Me3 and H3K27Ac binding as predicted by ENCODE, respectively. The boxed region was shown by ENCODE to bind MAX, MYC, ETS1, RNA PolR II, H3K4Me1, H3K4Me3 and H3K27Ac (https://genome.ucsc.edu). Eight cell types were tracked in the image. Light blue indicates HUVEC cells, while dark color indicates H7-ES cells. Scale = 1 kb. (G) Testing bidirectional *lncEGFL7OS* promoter. *LncEGF7OS* promoter was fused to a promoter-less Luciferase vector in forward (F) and reverse (R) directions, and tested for Luciferase activity with or without co-transfection of ETS1 or ETS1 mutant expression plasmid in 293 T cells. Shown here is the representative results from three repeats. (H) qRT-PCR showing that silencing of ETS1/2 result in the downregulation of lncEGFL7OS and pri-miR-126 (n = 3). *p<0.05; ***p<0.001.

DOI: https://doi.org/10.7554/eLife.40470.007

The following source data and figure supplements are available for figure 2:

**Source data 1.** *Figure 2* source data.

DOI: https://doi.org/10.7554/eLife.40470.011

**Figure supplement 1.** lncEGFL7OS RACE-PCR data and Real-time PCR data of ANP, PECAM1, EGFL7 and miR-126 in different tissues.

DOI: https://doi.org/10.7554/eLife.40470.008

**Figure supplement 1—source data 1.** *Figure 2—figure supplement 1* source data.

DOI: https://doi.org/10.7554/eLife.40470.009

**Figure supplement 2.** ChIP assay showing binding for the indicated factors to the promoter region (n = 3 each).

DOI: https://doi.org/10.7554/eLife.40470.010

patients (*Figure 2E*). Interestingly, the expression of EGFL7B and EGFL7C, as well as pri-miR-126, was also significantly upregulated in the hearts of DCM patients.

## Regulation of lncEGFL7OS expression by ETS factors through a bidirectional promoter in HUVECs

To dissect the lncEGFL7OS regulation mechanism in relation to its host gene EGFL7/miR-126, we aimed to identify the potential regulatory elements for lncEGFL7OS. We have analyzed the cell type-specific active element of the locus from online database UCSC genome browser (*Figure 2F*). A critical regulatory element is located on EGFL7B promoter between lncEGFL7OS and EGFL7/miR-126. Bioinformatics data from ENCODE indicate that *LncEGFL7OS* DNA contains a region positive for epigenetic marks including histone H3 trimethylated lysine four methylation (H3K4Me1) and H3K27Ac (mark poised and active enhancers), H3K4Me3 (marks promoter of protein coding genes), and binding sites for transcription factors MAX, MYC and RNA Polymerase (PolR) II. Several binding sites for ETS transcription factors were found in region. We have shown that its homologous region drives the EC-enriched LacZ reporter gene expression in mice (*Wang et al., 2008a*). Consistently, chromatin immunoprecipitation (ChIP) PCR assay using antibodies against MAX/MYC, RNA Pol II and histone H3 trimethylated lysine 4 (H3K4me3) demonstrated the binding of these factors specifically to the region but not a non-relevant nearby region, indicating that this region is transcriptionally active (*Figure 2—figure supplement 2A*). Additional potential promoters were not found in the region between lncEGFL7OS and EGFL7 transcripts by bioinformatics approach. Instead, CpG islands were found in the region. CpG islands in mammalian promoter regions tend to show bidirectional promoter activity (*Antequera, 2003*). Bidirectional promoters have been proposed to drive head-to-head gene transcription involving ncRNAs (*Uesaka et al., 2014*). Based on these, we tested a novel hypothesis that a bidirectional promoter (*lncEGFL7OS/EGFL7/miR-126* promoter) regulated by ETS factors drives the expression of both lncEGFL7OS and EGFL7/miR-126 in human ECs. The putative *lncEGFL7OS* promoter was cloned into a promoter-less luciferase reporter construct in either sense or anti-sense direction. By luciferase assay, the promoter in either direction exhibited similar activity under baseline in 293 T cells (*Figure 2G*). Moreover, ETS1 transcription factor significantly activated the promoter activity in either direction, while the ETS1mut that lacks the DNA-binding domain showed significantly reduced activation of the promoter (*Wang et al., 2008a*). ETS factors have been shown to regulate miR-126 expression in ECs (*Harris et al., 2010*).To further test

whether ETS factors are required to regulate lncEGFL7OS expression, ETS1 and ETS2 genes were silenced in HUVEC cells, and lncEGFL7OS and pri-miR-126 expression were examined by qRT-PCR. Both genes were significantly reduced by ETS1/2 silencing, suggesting ETS factors control the expression of both lncEGFL7OS and EGFL7/miR-126 (*Figure 2H*).

## Regulation of angiogenesis by lncEGFL7OS in vitro and in vivo

To define the potential role for lncEGFL7OS in angiogenesis, we performed EC-fibroblast co-culture assays after silencing lncEGFL7OS using two independent siRNAs in HUVEC cells (*Hetheridge et al., 2011*) (*Figure 3—figure supplement 1A–B*). When ECs are cultured on the top of a confluent fibroblast cell layer, ECs will proliferate to form 'islands' of ECs, and then sprout to form three-dimensional vascular tubules resembling capillaries which can be visualized by immunostaining with an antibody to EC-enriched human PECAM-1 (*Figure 3A*). Of note, control siRNA has a mild but not significant effect in angiogenesis in this model. Compared to the control siRNA, si-lincEGFL7#1 or si-lncEGFL7OS#2 significantly repressed the formation of vascular tubules at 7 days after co-culture as shown by PECAM-1 staining and the subsequent quantification of the vascular tube length (*Figure 3A–B*). Taken together, we conclude that lncEGFL7OS is required for proper angiogenesis in vitro.

To examine the requirement of lncEGFL7OS in vasculogenesis/angiogenesis in vivo, si-lncEGFL7OS or control transfected HUVEC cells were mixed with Matrigel and injected subcutaneously on the back midline of nude mice, and the primary vascular network was stained with antibody against human PECAM-1 at 14 days after Matrigel implantation. Compared to the well-connected vessel structure in the controls, fewer networking was observed in the lncEGFL7OS-silenced EC group (*Figure 3C–D*). Red blood cells and smooth muscle cells recruiting was detected in the formed vessels as proved by co-staining of human PECAM-1 and mouse Ter-119 (red blood cell marker) or mouse α-SMA (smooth muscle marker) staining, which suggests functionality of the vessels (*Figure 3C* and *Figure 3—figure supplement 2A–B*). These results indicate that lncEGFL7OS is required for proper angiogenesis in vivo.

To directly test the function of lncEGFL7OS in angiogenesis in human tissues, we developed a unique human choroid sprouting assay based on a previous publication (*Shao et al., 2013*). Briefly, human choroids were dissected from the donor eyes from the eye bank, and were cut into approximately 4 mm$^2$ pieces and transfected with control or lncEGFL7OS siRNAs overnight. The choroids were then seeded in the Matrigel and cultured in EGM-2 medium for up to 10 days. Silencing of lncEGFL7OS by siRNAs (a mix of siRNA #1 and 2 at half concentration used for other assays) in the system was confirmed by qRT-PCR (*Figure 3E*). In the control choroid, significant sprouting was observed at day 10 with an average distance of ~1200 μm (*Figure 3F*). Compared to the control, lncEGFL7OS siRNAs drastically repressed human choroid sprouting, establishing a critical role for lncEGFL7OS in angiogenesis in human tissues (*Figure 3F–G*). The EC identity of the sprouting cells was confirmed by ICAM-2 and isolectin B4 co-staining (*Figure 3H*).

## Regulation of EC proliferation and migration by lncEGFL7OS in vitro

To dissect the cellular mechanism whereby lncEGFL7OS regulates angiogenesis, a BrDU incorporation assay was carried out to analyze EC proliferation upon lncEGFL7OS silencing. Under starvation condition, si-lincEGFL7#2 significantly decreased EC proliferation as shown by BrdU incorporation compared to the random control, while the effect from si-lncEGFL7OS#1 was not statistically significant (*Figure 4A*). However, the EC proliferation induced by VEGF treatment was significantly repressed by either si-lncEGFL7OS#1 or si-lncEGFL7OS#2. To further characterize the reduced EC proliferation after lncEGFL7OS knockdown, the cell cycle profile was quantified after flow cytometry under normal culture conditions. A significant increase in the percentage of cells in the G0/G1 phase was observed upon lncEGFL7OS knockdown (*Figure 4B–C*). Accordingly, cells in the S and G2/M phase are significantly decreased. This indicates a G1 arrest in the si-lncEGFL7OS treated cells. We also determined whether EC migration is affected by lncEGFL7OS knockdown. Using a scratch wound assay, we found that compared to the control, lncEGFL7OS silencing significantly repressed EC migration in response to VEGF treatment after wound scratch (*Figure 4D–E*). To assess whether lncEGFL7OS silencing results in EC death, TUNEL assay was performed. In the control condition,~0.4% of EC cells undergo cell death, silencing of lncEGFL7OS by siRNA#1 and #2

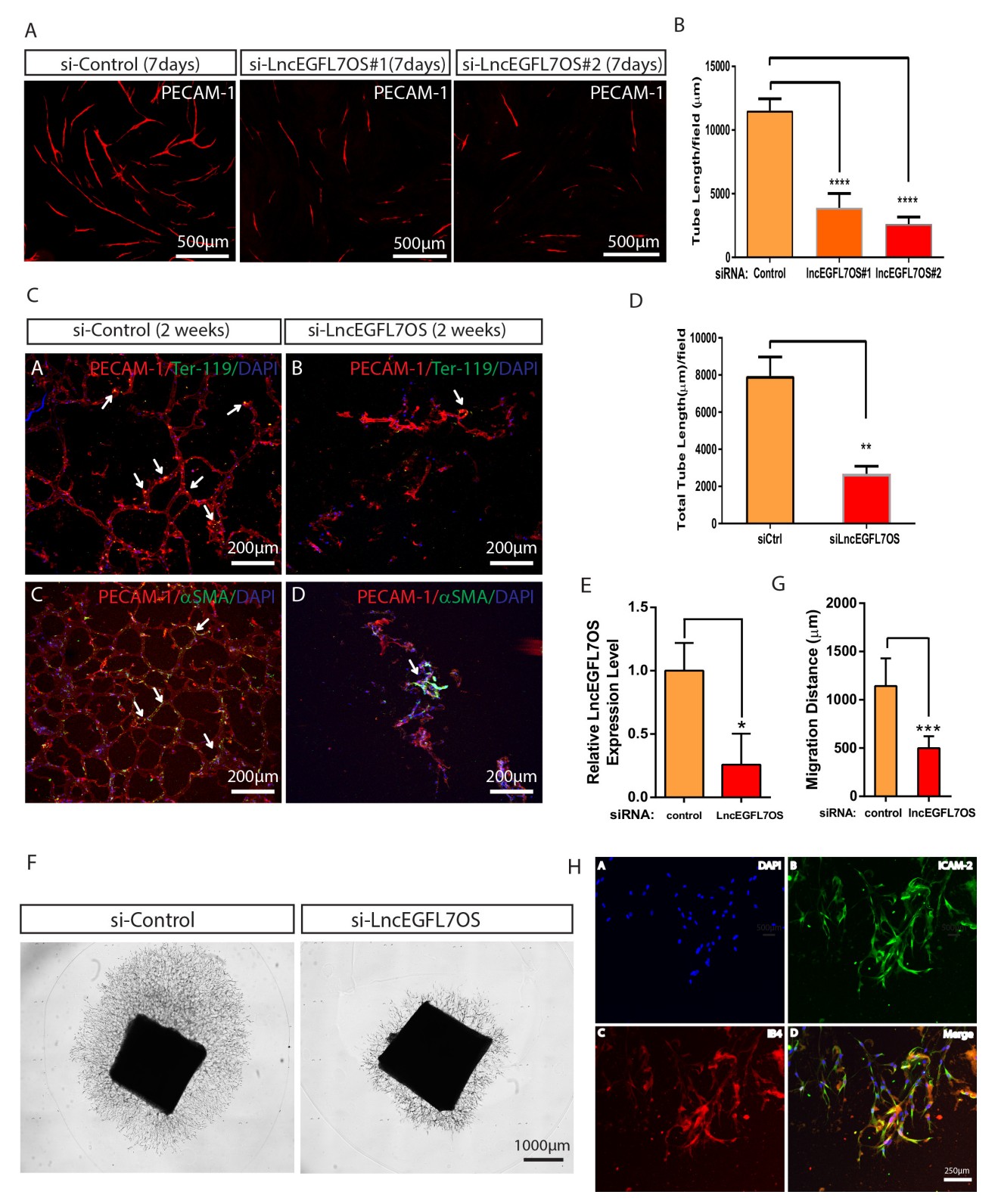

**Figure 3.** Regulation of angiogenesis by lncEGFL7OS in vitro, ex vivo and in vivo. (**A**) Decreased capillary tube formation at 7 days after lncEGFL7OS silencing in HUVECs in an EC-fibroblast co-culture assay. The capillaries are stained with PECAM-1 antibody. Scale bar equals to 500 μm. (**B**) Quantification of total tube length in A (n = 3 each). Two independent lncEGFL7OS siRNAs were used for quantification. ****p<0.0001. (**C**) Defective EC networking at 14 days after lncEGFL7OS silencing in an in vivo Matrigel implantation model. A mix of si-linEGFL7OS#1 and si-lncEGFL7OS#2 was

*Figure 3 continued on next page*

*Figure 3 continued*

used for the experiments. HUVEC cells in the Matrigel were stained with human PECAM-1 antibody (Red), mouse red blood cells were stained with mouse Ter-119 (Green) antibody and mouse smooth muscle cells were stained with α-SMA (Green) antibody. Arrows label the representative areas with overlapping staining in the Matrigel. DAPI was used to stain nucleus. Scale bar equals to 200 μm. (D) Quantification of tubule length in C (n = 3 mice each). **p<0.01. (E) Inhibition of lncEGFL7OS expression by si-lncEGFL7OS-1/2 in human choroids cultured ex vivo, as revealed by qRT-PCR. (n = 3) (F) Representative picture of human choroid sprouting angiogenesis after lncEGFL7OS knockdown; G) Quantification of choroid sprouting distance in F. (n = 6) (H) Representative ICAM2 (green) and Isolectin B4 (red) staining of the choroid sprouts in F. Scale bar equals to 250 μm.
DOI: https://doi.org/10.7554/eLife.40470.012

The following source data and figure supplements are available for figure 3:

**Source data 1.** *Figure 3* source data.
DOI: https://doi.org/10.7554/eLife.40470.016

**Figure supplement 1.** Schematics of lncEGFL7OS siRNA localization and siRNA knockdown efficiency.
DOI: https://doi.org/10.7554/eLife.40470.013

**Figure supplement 1—source data 1.** *Figure 3* source data.
DOI: https://doi.org/10.7554/eLife.40470.014

**Figure supplement 2.** Immunostaining of the Matrigel assay in *Figure 3C*.
DOI: https://doi.org/10.7554/eLife.40470.015

significantly increased EC death to ~0.55% and~0.64%, respectively (*Figure 4—figure supplement 1*). Therefore, the increase of EC death by si-lncEGFL7OS is statistically significant, but probably not biologically important with regard to the angiogenic phenotypes observed. These results indicate that lncEGFL7OS is required for proper EC proliferation and migration in vitro.

## Overexpression of lncEGFL7OS enhances angiogenesis in an EC/Fibroblast co-culture angiogenesis model

We further examined whether overexpression of lncEGFL7OS in ECs enhances angiogenesis. To do so, lncEGFL7OS or control LacZ adenoviruses were generated, and used to infect HUVEC cells at multiplicity of infection at 50. Infected ECs were cultured on a fibroblast mono layer, and their angiogenic response was examined by staining with an antibody to PECAM-1 at 7 days after co-culture. The efficiency of the virus was verified by qRT-PCR. Over 2000-fold lncEGFL7OS was achieved in ECs after virus infection (*Figure 4—figure supplement 2A*). No significant differences were observed in Ad-lacZ infected samples compared to noninfection controls. LncEGFL7OS overexpression enhanced angiogenesis as shown by the significantly increased total tube length compared to the controls (*Figure 4—figure supplement 2B–C*). These data indicate that overexpression of lncEGFL7OS is sufficient to enhance EC angiogenesis.

## Regulation of EGFL7/miR-126 expression by lncEGFL7OS

lncRNAs could exert regulatory function in cis on the neighboring genes in the nucleus (*Ørom et al., 2010*). Since lncEGFL7OS is located in the opposite strand neighboring EGFL7/miR-126, we surmised that lncEGFL7OS regulates angiogenesis by controlling EGFL7/miR-126 expression. The expression of EGFL7B-C and miR-126 was examined by qRT-PCR upon lncEGFL7OS knockdown. As shown in *Figure 5A*, EGFL7B and C expression was dramatically decreased upon lncEGFL7OS knockdown. The downregulation of EGFL7 at protein level by lncEGFL7OS knockdown was confirmed by Western blot analysis (*Figure 5—figure supplement 1*). Similarly, the expression of both miR-126 and miR-126*, a microRNA located in the intron of EGFL7 gene, is also downregulated by lncEGFL7OS knockdown (*Figure 5B*). miR-126 has been shown to modulate MAP kinase signaling and PI3K-AKT signaling by targeting Spred-1 and PI3KR2, respectively (*Fish et al., 2008*; *Wang et al., 2008a*; *Kuhnert et al., 2008*). Consistent with the downregulation of miR-126, phosphorylation of ERK1/2 and AKT induced by VEGF was significantly reduced in ECs transfected with si-lncEGFL7OS#1 or si-lncEGFL7OS#2 compared to the controls (*Figure 5C*). We also examined whether lncEGFL7OS overexpression increases the expression of EGFL7 and miR-126. As expected, a ~ 2 fold upregulation of miR-126 and a ~ 3 fold increase of EGFL7B were observed when lncEGFL7OS is overexpressed in ECs (*Figure 5—figure supplement 2A–B*). To determine whether EGFL7 and miR-126 can mediate the angiogenic response of lncEGFL7OS, we tested the capability of miR-126 expressing adenovirus and EGFL7 protein in rescuing the anti-angiogenic phenotype of si-

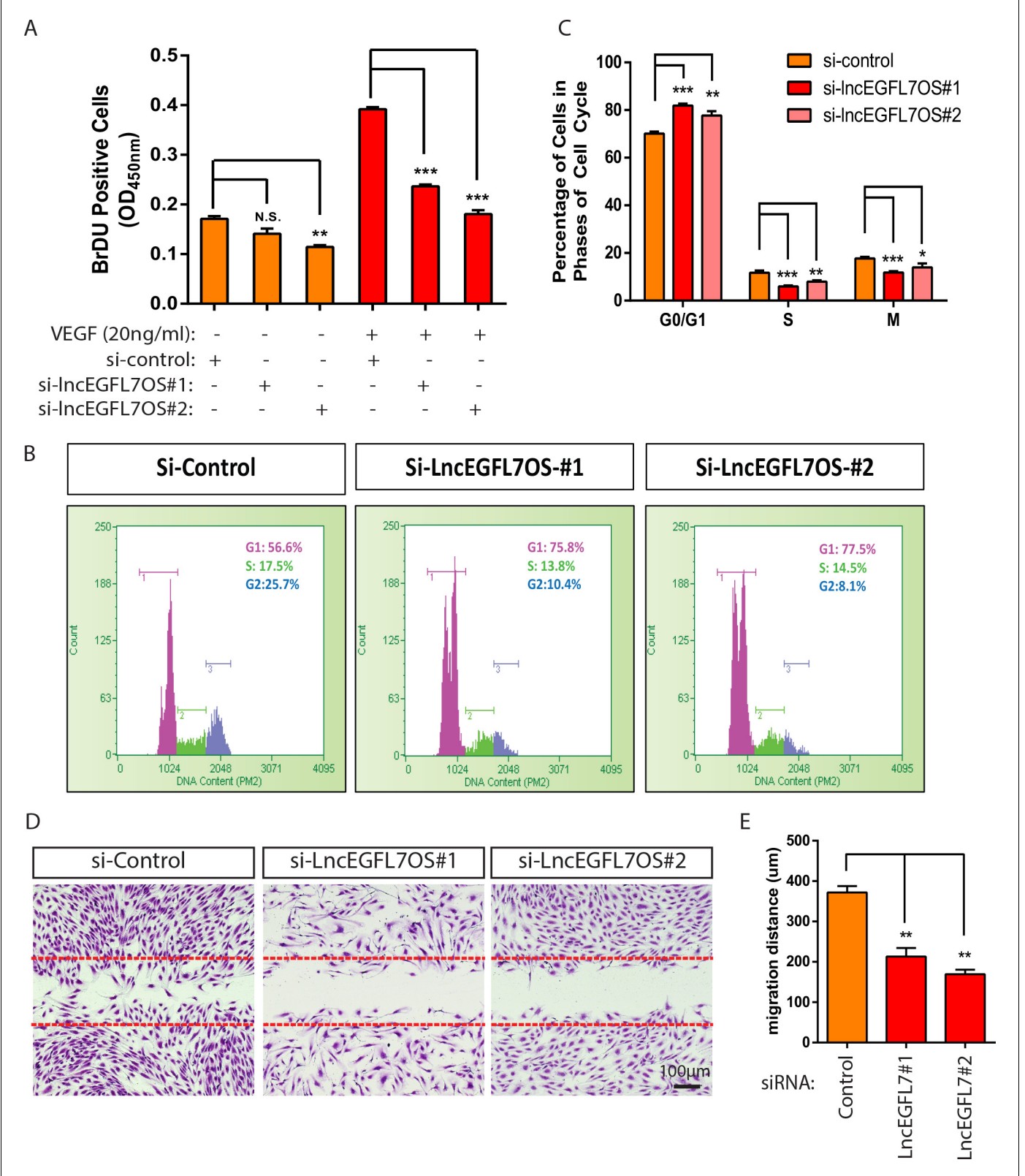

**Figure 4.** Regulation of EC proliferation and migration by lncEGFL7OS. (**A**) Quantification of EC proliferation in response to VEGF-A as indicated by BrDU incorporation after lncEGFL7OS silencing (n = 3). (**B**) Representative ell cycle profile in ECs after lncEGFL7OS silencing. (**C**) Statistics of the percentage of cells in different phases of cell cycle after lncEGFL7 silencing. (n = 3) *p<0.05; **p<0.01; ***p<0.001. (**D**) Repression of cell migration in a

*Figure 4 continued on next page*

*Figure 4 continued*

scratch wound assay in ECs after lncEGFL7OS silencing. Dashed lines indicate the initial position of cells. Scale bar equals to 100 μm. (**E**) Quantification of EC migration in D (n = 3). **p<0.01.

DOI: https://doi.org/10.7554/eLife.40470.017

The following source data and figure supplements are available for figure 4:

**Source data 1.** *Figure 4* source data.

DOI: https://doi.org/10.7554/eLife.40470.022

**Figure supplement 1.** Quantification of TUNEL positive cells in ECs transfected siRNAs for lncEGFL7OS (n = 3).* p<0.05.

DOI: https://doi.org/10.7554/eLife.40470.018

**Figure supplement 1—source data 1.** *Figure 4—figure supplement 1* source data.

DOI: https://doi.org/10.7554/eLife.40470.019

**Figure supplement 2.** Effect of lncEGFL7OS oeverexpression in angiogenesis.

DOI: https://doi.org/10.7554/eLife.40470.020

**Figure supplement 2—source data 1.** *Figure 4—figure supplement 2* source data.

DOI: https://doi.org/10.7554/eLife.40470.021

lncEGFL7OS. The combination of miR-126 and EGFL7 enhanced angiogenesis in the wild-type HUVECs, and rescued the anti-angiogenic effect of lncEGFL7OS silencing to a great extent in an EC-Fibroblast cell co-culture model (*Figure 5D–E*). These results indicate that lncEGFL7OS is critical for maintaining maximal expression of EGLF7/miR-126, which is required for VEGF signaling and angiogenesis through MAPK and PI3K/AKT pathways.

## lncEGFL7OS regulates EGFL7/miR-126 promoter activity by interacting with MAX transcription factor

To study the mechanism whereby lncEGFL7OS regulates EGFL7/miR-126 expression, we hypothesized that lncEGFL7OS regulates EGFL7/miR-126 promoter/enhancer activity by interacting with MAX transcription factor. MAX was predicted as one of the top lncEGFL7OS-interacting proteins by lncRNA interaction prediction program catRAPID (*Bellucci et al., 2011*). Online database UCSC genome browser predicts the existence of MAX binding sites between lncEGFL7OS and EGFL7/ miR-126 genes (*Figure 6A*). We first tested whether lncEGFL7OS interacts with MAX protein in ECs. RNA immunoprecipitation (RIP) assays showed that lncEGFL7OS RNA was pulled down in the nuclear lysate by a Chip-grade antibody to MAX, and this interaction was increased by lncEGFL7OS overexpression (*Figure 6B*). To dissect the domains in lncEGFL7OS that interact with MAX, lncEG-FL7OS was separated into three fragments according to the predicted secondary structure (*Figure 6C*). Three different fragments (F1 to F3) were cloned into expression vectors, and transfected into RPE cells that have undetectable endogenous lncEGFL7OS expression. Similar RIP RT-PCR assays demonstrated that F1 fragment in the 5' end of lncEGFL7OS is the major domain that interacts with MAX protein (*Figure 6D*).

We further examined whether MAX protein binds to the bidirectional lncEGFL7OS/EGFL7/miR-126 promoter/enhancer. ChIP-PCR assays confirmed the specific binding of MAX to this region in ECs (*Figure 6E*). Moreover, overexpression of lncEGFL7OS significantly increased MAX binding to this region. As control, MAX protein was not enriched in a non-relevant control DNA region (*Figure 6—figure supplement 1A*). MAX has been shown to dimerize with MYC and stimulate histone acetylation and gene transcription (*Vervoorts et al., 2003*). Our co-immunoprecipitation assay confirmed the interaction of MAX with p300, a component in the p300/CBP co-activator complex that has intrinsic histone acetyltransferase activities, in ECs (*Figure 6—figure supplement 1B*). We therefore determined whether acetylated H3K27 (H3K27ac), a marker for active enhancer, is enriched in this region, and found H3K27ac was indeed enriched in the region, and this enrichment was further increased by lncEGFL7OS overexpression (*Figure 6F*). To confirm whether the interaction of lncEG-FL7OS with MAX is required for angiogenesis, lncEGFL7OS-F(2 + 3) that does not contain the F1 region was cloned and used to make adenovirus. Overexpression of lncEGFL7OS-F(2 + 3) by adenovirus neither affected EGFL7B and miR-126 expression, nor impacted angiogenesis in an EC-fibroblast co-culture assay (*Figure 6—figure supplement 1C–F*), suggesting the requirement of lncEGFL7OS/MAX interaction in angiogenesis. Together, these results suggest that lncEGFL7OS

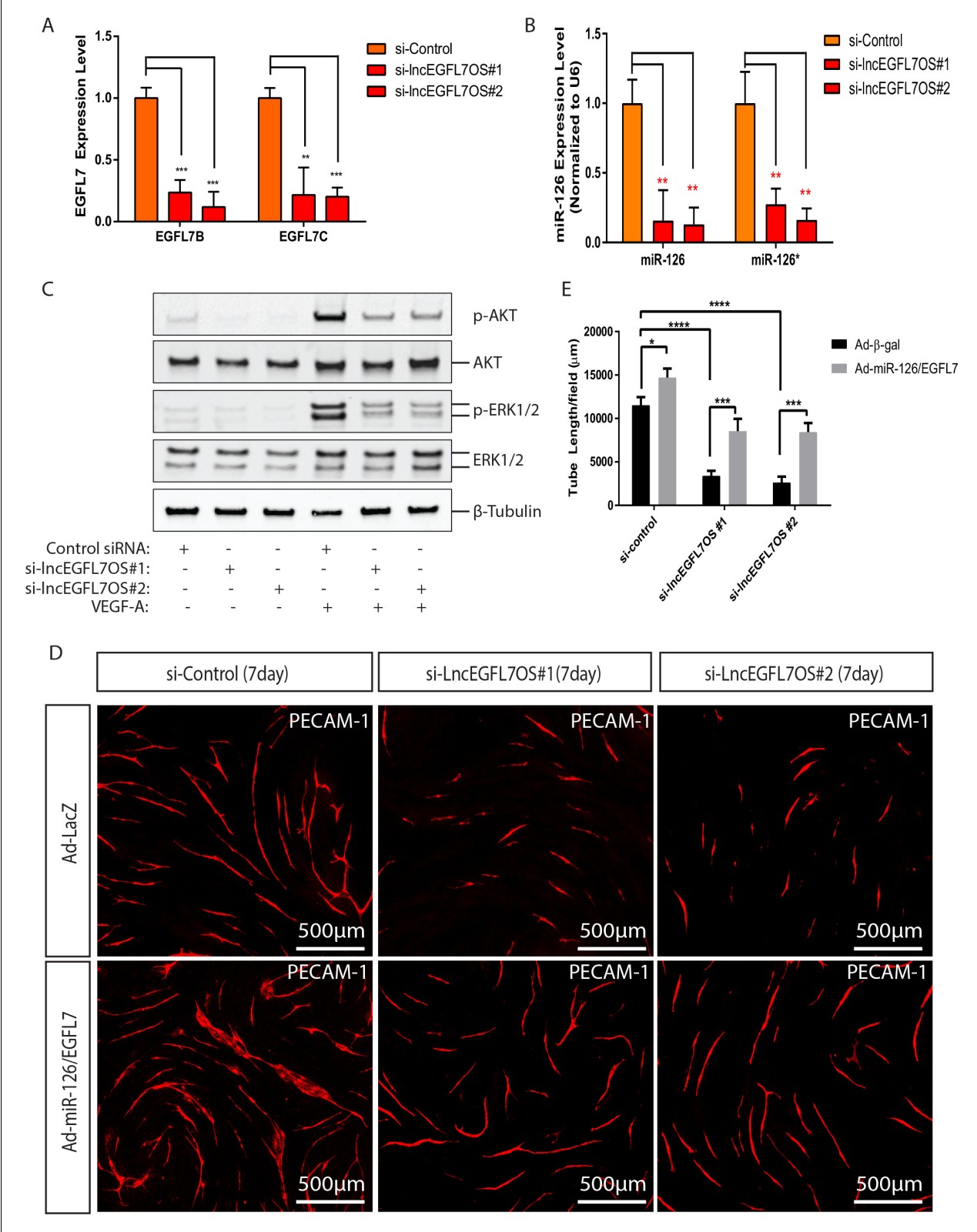

**Figure 5.** Regulation of EGFL7/miR-126 and angiogenic signaling by lncEGFL7OS. (A) Expression of *EGFL7 B* and *EGFL7C* by qRT-PCR after lncEGFL7OS knockdown in ECs (n = 3). GAPDH served as normalization control. (B) Expression of miR-126 and miR-126* after lncEGFL7OS knockdown in ECs (n = 3). U6 served as normalization control. (C) Regulation of ERK1/2 and AKT phosphorylation by lncEGFL7OS knockdown in ECs in response to VEGF treatment, as revealed by Western blot. Total ERK1/2 and AKT were used as controls. β-Tubulin was used as a loading control. (D) Rescue of the

*Figure 5 continued on next page*

Figure 5 continued

lncEGFL7OS-knockdown angiogenic phenotype by EGFL7 protein/Adeno-miR-126 in an EC-fibroblast co-culture assay. Scale bar equals to 500 µm. (E) Quantification of the total tube length in D (n = 3). *p<0.05. ***p<0.001, ****p<0.0001.

DOI: https://doi.org/10.7554/eLife.40470.023

The following source data and figure supplements are available for figure 5:

**Source data 1.** *Figure 5* source data.

DOI: https://doi.org/10.7554/eLife.40470.027

**Figure supplement 1.** Expression of EGFL7 protein by Western blot after lncEGFL7OS knockdown in ECs (n = 3).

DOI: https://doi.org/10.7554/eLife.40470.024

**Figure supplement 2.** Upregulation of miR-126 and EGFL7 by lncEGFL7OS oeverexpression.

DOI: https://doi.org/10.7554/eLife.40470.025

**Figure supplement 2—source data 1.** *Figure 5—figure supplement 2* source data.

DOI: https://doi.org/10.7554/eLife.40470.026

promotes the binding of MAX protein to the bidirectional promoter/enhancer region of lncEG-FL7OS/EGFL7/miR-126, and enhances their transcription, and therefore angiogenesis.

To examine whether MAX is required for regulating lncEGFL7OS/EGFL7/miR-126 expression, two specific siRNAs were used to silence MAX expression (*Figure 6G*). MAX silencing resulted in significantly decreased expression of EGFL7, lncEGFL7OS and miR-126 (*Figure 6H–J*). Consistently, MAX silencing led to repressed angiogenesis as shown by EC-Fibroblast co-culture assays (*Figure 6K*). We further determine whether MAX silencing overrides the increased expression of miR-126 induced by adenovirus expressing lncEGFL7OS. As shown in *Figure 6L*, the induction of miR-126 expression by lncEGFL7OS overexpression was blunted by MAX knockdown. To determine whether lncEG-FL7OS is required for MAX recruiting to the EGFL7/miR-126 promoter/enhancer, similar ChIP-PCR was performed after lncEGFL7OS knockdown. As shown in *Figure 7A–B*, silencing of lncEGFL7OS significantly reduced MAX binding to the EGFL7/miR-126 promoter/enhancer as well as H3K27 acetylation at the locus. Together, our data indicate that lncEGFL7OS regulates EGFL7/miR-126 expression by interaction with MAX transcription factor, which enhances H3K27 acetylation in the lncEGFL7OS/EGFL7/miR-126 enhancer/promoter region.

Since lncEGFL7OS interacts with MAX, we asked whether other known MAX target genes, including Cyclin D2 and DHFR, are regulated by lncEGFL7OS (*Mai and Jalava, 1994*; *Bouchard et al., 2001*). These two genes were confirmed to be MAX targets in ECs by siRNA experiments and ChIP assays (*Figure 7C–D and G–H*). Overexpression of lncEGFL7OS enhanced the expression of Cyclin D2 and DHFR (*Figure 7E–F*), which could be explained by the increased binding of MAX and increased H3K27 acetylation at their respective promoters (*Figure 7G–J*). However, neither Cyclin D2 nor DHFR expression was repressed by lncEGFL7OS knockdown (*Figure 7K–L*). These data suggest that, although lncEGFL7OS is capable of regulating other MAX target genes when overexpressed, lncEGF7OS does not act in trans to regulate angiogenesis through MAX under normal conditions.

## Inhibition of angiogenesis by CRISPR-mediated targeting of the EGFL7/miR-126/lncEGFL7OS locus

To further study the regulatory mechanism and the therapeutic targeting potential of the EGFL7/miR-126/lncEGFL7OS locus, a dCas9-KRAB system, in which a catalytically inactive Cas9 is fused to KRAB transcriptional repressor, was utilized to test the effect of silencing this locus on angiogenesis (*Qi et al., 2013*). Two guide RNAs (sgRNAs), with one targeting the genomic region between the EGFL7B and lncEGFL7OS transcription start sites and the other targeting the lncEGFL7OS intron region, were designed to guide sequence-specific transcription repression mediated by dCas9-KRAB (*Figure 8A*). By EC-fibroblast co-culture assay, lentivirus expressing sgRNA-1 or sgRNA-2 significantly repressed EC angiogenesis only when dCas9-KRAB was co-expressed (*Figure 8B–C*). Of note, Lenti-dCas9-KRAB alone did not significantly impact angiogenesis, ruling out the potential side-effects of dCas9-KRAB overexpression. When gene expression near this locus was examined, the expression of EGFL7B, miR-126 and lncEGFL7OS was drastically repressed by sgRNA-1, and to a less extent by sgRNA-2 (*Figure 8D*). These data support the co-regulation of EGFL7/miR-126 and

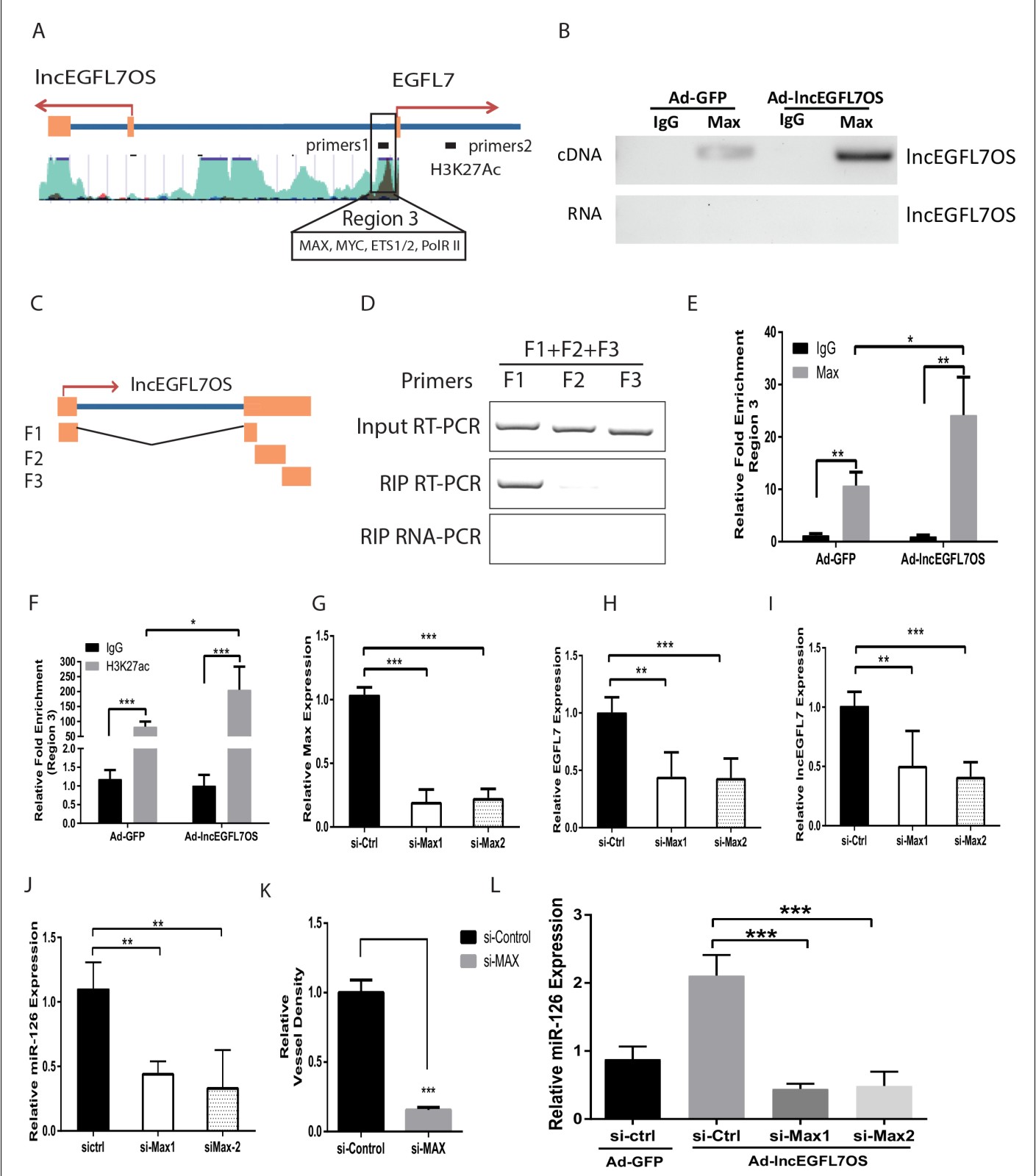

**Figure 6.** lncEGFL7OS regulates EGFL7/miR-126 transcription by interaction with MAX transcription factor. (**A**) Schematic EGFL7/miR-126 enhancer/ promoter region. The boxed region is predicted by ENCODE to bind MAX and H3K27Ac. (**B**) RIP-PCR showing binding of MAX to lncEGFL7OS in ECs. Overexpression of lncEGFL7OS by adenovirus enhances MAX binding. The bottom line shows a non-RT control for PCR. (**C**) Schematics of the lncEGFL7OS fragments for the MAX-binding assay. (**D**) RIP PCR showing specific binding of F1 fragment of lncEGFL7OS to MAX protein. Input RT-PCR

*Figure 6 continued on next page*

*Figure 6 continued*

showed the expression of lncEGFL7 fragments in transfected RPE-19 cells. RIP RT-PCR showed the specific binding of F1 fragment to MAX by RIP assay. RIP RNA-PCR showed the DNase I treated non reverse transcription control. (E) ChIP-PCR showing specific binding of MAX to region three in A. Overexpression of lncEGFL7OS enhances MAX binding to the region. *p<0.05; **p<0.01. (F) ChIP-PCR showing specific binding of H3K27ac to region three in A. Overexpression of lncEGFL7OS enhances H3K27ac binding to the region. *p<0.05; ***p<0.001. (G) Silencing of MAX expression by two independent siRNAs as shown by qRT-PCR. ***p<0.001. (H) Downregulation of EGFL7B by MAX silencing in ECs. **p<0.01, ***p<0.001. (I) Downregulation of lncEGFL7OS by MAX silencing in ECs. **p<0.01, ***p<0.001. (J) Downregulation of miR-126 by MAX silencing in ECs. **p<0.01. (K) Quantification of vessel density in an EC-Fibroblast co-culture assay after MAX silencing. A mix of two independent MAX siRNAs was used in the assay. **p<0.01. (L) MAX silencing blunts the induction of miR-126 by lncEGFL7OS-expressing adenovirus. ***p<0.001.

DOI: https://doi.org/10.7554/eLife.40470.028

The following source data and figure supplements are available for figure 6:

**Source data 1.** *Figure 6* source data.
DOI: https://doi.org/10.7554/eLife.40470.031
**Figure supplement 1.** Overexpression of lncEGFL7OS(F2+3) does not affect miR-126 and EGFL7B expression.
DOI: https://doi.org/10.7554/eLife.40470.029
**Figure supplement 1—source data 1.** *Figure 6—figure supplement 1* source data.
DOI: https://doi.org/10.7554/eLife.40470.030

lncEGFL7OS in the locus, and suggest the potential of therapeutic targeting angiogenesis by simultaneously targeting these three genes using a CRISPR-mediated approach.

## Discussion

In this study, we have identified ~500 EC-restricted lncRNAs by comparing the lncRNA/mRNA profile from EC and non-EC lines. The EC- or vasculature-restriction of a list of candidate lncRNAs was confirmed by qRT-PCR and bioinformatics approaches. We further reported a human/primate-specific EC-enriched *lncEGFL7OS* that is located in the opposite strand neighboring the *EGFL7/miR-126* gene. Expression of lncEGFL7OS in ECs is regulated by ETS transcription factors through a bidirectional promoter. Silencing of lncEGFL7OS represses EC proliferation and migration, therefore impairing angiogenesis in vitro and human choroid sprouting angiogenesis ex vivo; while overexpression of lncEGFL7OS enhances angiogenesis in ECs. Moreover, CRISPR-mediated targeting of EGLF7/miR-126/lncEGFL7OS locus inhibited angiogenesis, suggesting therapeutic potential of targeting this locus. Upregulation of lncEGFL7OS and EGFL7/miR-126 was observed in the hearts of DCM patients, which may reflect the compensatory vascularization/angiogenesis in DCM. Mechanistically, lncEGFL7OS regulates angiogenic signaling through enhancing EGFL7/miR-126 transcription by interaction with MAX transcription factor, which regulates EGLF7/miR-126/lncEGFL7OS promoter activity (*Figure 9*).

### Identification of EC-restricted lncRNAs

Our data areconsistent with a recent publication that identified EC-restricted lncRNAs (*Man et al., 2018*). Several lncRNAs, including lncEGFL7OS, HHIP-AS1 and SENCR, were in the short list from both microarrays. The difference from our results may reflect the different cell types used in the microarrays. We found 498 lncRNAs are enriched in three different primary EC lines compared to non-EC lines using a cutoff of 2. By hierarchical cluster analysis, lncRNA-based clustering appeared to be a stronger classifier for EC lines than mRNA clustering. This is consistent with the general perception that lncRNAs exhibit better tissue specificity than mRNAs (*Derrien et al., 2012*). We also found significant variability in lncRNA expression among EC lines, consistent the observed heterogeneity among ECs. Given the central importance of ECs in vascular biology, this dataset may provide a foundation to study the regulation and function for lncRNAs in various vascular development and disease models. Of note, we also found many lncRNAs are highly expressed in ECs, but those lncRNAs are not necessarily EC-specific (data not shown). Those lncRNAs may also important function in cell types including ECs.

Looking deep into the gene list, 91 lncRNAs of the 498 EC-restricted genes have protein coding genes within 10 kb, and about a third of them showed parallel or inverse expression pattern to the associated genes. Functional enrichment analysis indicates that EC-restricted lncRNAs are associated

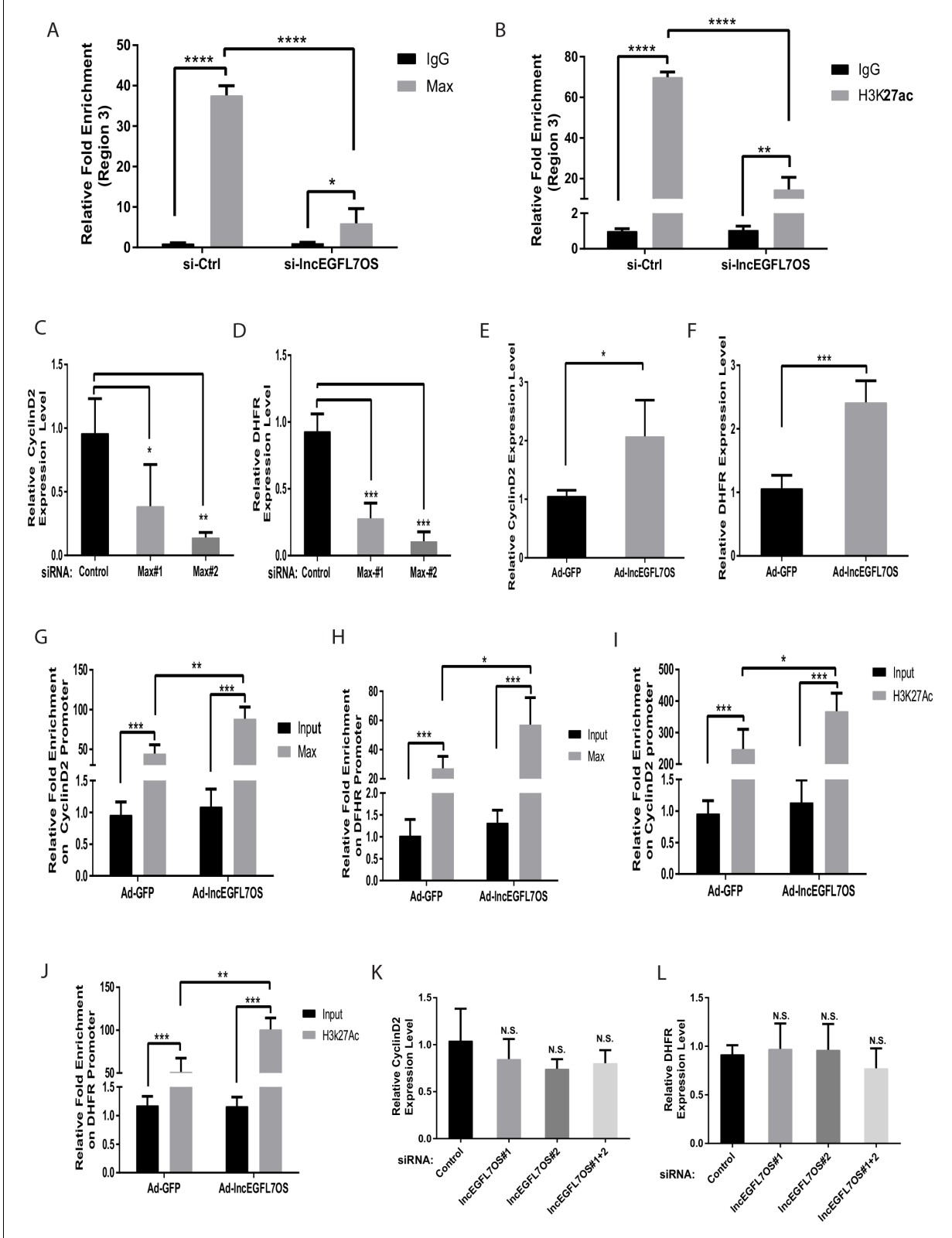

**Figure 7.** lncEGFL7OS-dependent MAX-regulated gene expression is locus dependent. (**A**) ChIP-PCR showing specific binding of MAX to region 3 (as in *Figure 6A*). Silencing of lncEGFL7OS decreased MAX binding to the region (n = 3). *p<0.05; ***p<0.001. (**B**) ChIP-PCR showing specific binding of H3K27ac to region 3 (as in *Figure 6A*). Silencing of lncEGFL7OS decreased H3K27ac binding to the region (n = 3). **p<0.01; ***p<0.001. (**C**) Repression of Cyclin D2 expression in HUVEC cells by MAX knockdown using two independent siRNAs (n = 3). *p<0.05; **p<0.01. (**D**) Repression of DHFR

*Figure 7 continued on next page*

*Figure 7 continued*

expression in HUVEC cells by MAX knockdown using two independent siRNAs (n = 3). *p<0.05; **p<0.01. (E) lncEGFL7OS overexpression enhances Cyclin D2 expression (n = 3). *p<0.05. (F) lncEGFL7OS overexpression enhances DHFR expression (n = 3). ***p<0.001. (G) ChIP-PCR showing specific binding of MAX to the Cyclin D2 promoter. Overexpression of lncEGFL7OS further enhances MAX binding to the region (n = 3). **p<0.01; ***p<0.001. (H) ChIP-PCR showing specific binding of MAX to the DHFR promoter. Overexpression of lncEGFL7OS further enhances MAX binding to the region (n = 3). *p<0.05; ***p<0.001. (I) ChIP-PCR showing increased H3K27 acetylation at the Cyclin D2 promoter. Overexpression of lncEGFL7OS further enhances H3K27 acetylation at the region (n = 3). *p<0.05; ***p<0.001. (J) ChIP-PCR showing increased H3K27 acetylation at the DHFR promoter. Overexpression of lncEGFL7OS further enhances H3K27 acetylation at the region (n = 3). **p<0.01; ***p<0.001. (K) qRT-PCR showing no effect of lncEGFL7 knockdown on Cyclin D2 expression (n = 3). N.S., non-significant. (L) qRT-PCR showing no effect of lncEGFL7 knockdown on DHFR expression (n = 3). N.S., non-significant.

DOI: https://doi.org/10.7554/eLife.40470.032

The following source data is available for figure 7:

**Source data 1.** *Figure 7* source data.

DOI: https://doi.org/10.7554/eLife.40470.033

with genes involved in vascular development. Those lncRNAs may be good candidates for further functional studies.

## Evolution of lncEGFL7OS/EGFL7/miR-126 locus

The evolution of *EGFL7/miR-126* locus exemplifies the evolution of the vascular system. EGFL7 encodes an EGF-like domain containing protein that is specifically secreted by vascular ECs (*Parker et al., 2004*). It is conserved among vertebrates but an orthologue is also found in Drosophila melanogaster (CG7447) (*Nikolic et al., 2010*). miR-126 and miR-126* are encoded by the intron of EGFL7, and are conserved from Fugu in vertebrates to homo sapiens (*Wang et al., 2008a*). They are the only miRNAs that are known to be specifically in EC lineage and hematopoietic stem cells. Loss-of-function studies in mice and zebrafish revealed an important function of miR-126 in governing vascular integrity and angiogenesis (*Fish et al., 2008*; *Wang et al., 2008a*). $Egfl7^{-/-}$ mice display similar vascular abnormalities to $MiR126^{-/-}$ mice, including edema, defective cranial vessel and retinal vascularization (*Schmidt et al., 2007*). However, an independent study suggests that the vascular phenotype of $Egfl7^{-/-}$ mice could be attributed to the $MiR126$ deletion (or downregulation) in the mice (*Kuhnert et al., 2008*). The important regulatory function of miR-126 in vascular integrity and angiogenesis is correlated with its appearance during the evolution of vascular system in vertebrates. Besides, miR-126 also has documented functions in vascular inflammation, as well as innate and adaptive immunity (*Harris et al., 2008*; *Mattes et al., 2009*; *Agudo et al., 2014*). That also correlates with the evolutionary innovation of adaptive immune system in vertebrates. These support an important function of *EGFL7*/miR-126 locus from the evolutionary point of view. To further dissect the function and regulation of the locus during evolution from vertebrates to humans, we identified *lncEGFL7OS*, which is located in the opposite strand neighboring the EGFL7/miR-126 gene. It only exists in humans and several other primates, including rhesus monkeys, but not in other lower vertebrate species including mice. Although we showed significant function of lncEGFL7OS in human angiogenesis, the full spectrum of lncEGFL7OS function remains to be established.

## lncEGFL7OS is a human/primate-specific EC-restricted lncRNA required for proper human angiogenesis

The expression of lncEGFL7OS is restricted to ECs and highly vascularized tissues, which is consistent with the expression of its host genes EGFL7 and miR-126. As to its regulatory mechanisms, we found that both lncEGFL7OS and miR-126 are regulated by ETS1/2 factors in ECs through a bidirectional promoter. We found that lncEGFL7OS is required for proper angiogenesis in vitro by using EC-fibroblast co-culture vasculogenesis/angiogenesis assays. Conversely, overexpression of lncEGFL7OS enhances angiogenesis. Using a human choroid sprouting angiogenesis model we developed, we further demonstrated that lncEGFL7OS is required for human sprouting angiogenesis. This study indicates that three different transcripts from the EGFL7/miR-126 locus, including lncEGFL7OS, EGFL7 and miR-126, have important functions in angiogenesis. EGFL7 and miR-126 have been previously shown to regulate angiogenesis (*Nikolic et al., 2010*). EGFL7 is essential for vascular tube formation during vasculogenesis in zebrafish (*Parker et al., 2004*). The importance of miR-

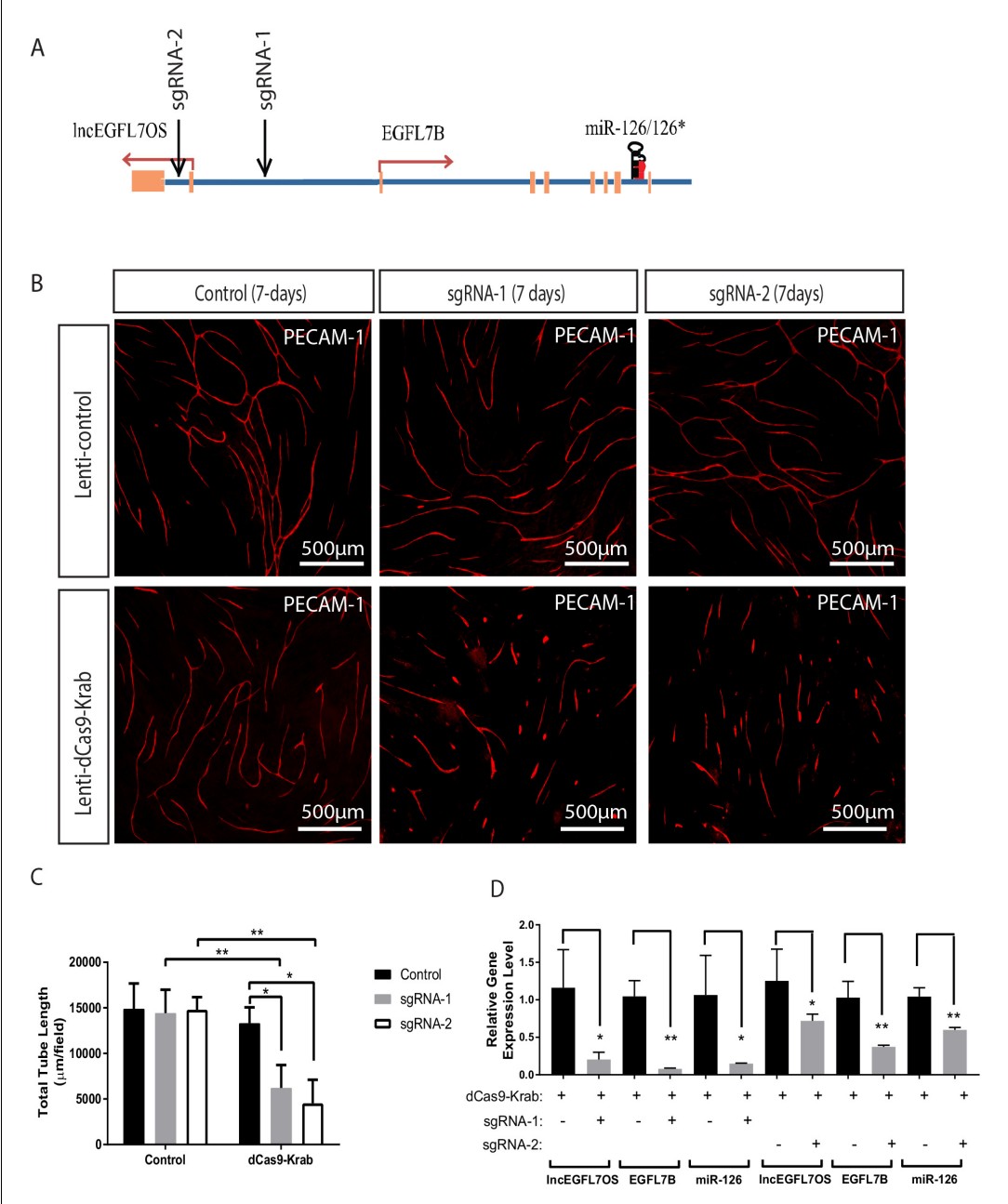

**Figure 8.** Inhibition of angiogenesis by CRISPR-mediated targeting of the EGFL7/miR-126/lncEGFL7OS locus. (**A**) Schematic locations of the sgRNAs in the EGFL7/miR-126/lncEGFL7OS genes. (**B**) Representative images showing sgRNA mediated repression of angiogenesis in an EC-fibroblast co-culture assay. The capillaries are stained with PECAM-1 antibody. Scale bar equals to 500 μm. All constructs were made into lentivirus. Lenti-control vector: pLJM1-EGFP; Lenti-dCas9-Krab: pHR-SFFV-dCas9-BFP-KRAB; sgRNA-1: lentiGuide-gRNA1; sgRNA-2: lentiGuide-gRNA2; Control: lentiGuide-Puro. (**C**) Quantification of total tube length in B (n = 3 each). Two independent sgRNAs were used for quantification. *p<0.05; **p<0.01. (**D**) Expression of lncEGFL7OS, EGFL7B and miR-126 at 48 hr after transduction in B by qRT-PCR. *p<0.05; **p<0.01.
DOI: https://doi.org/10.7554/eLife.40470.034
The following source data is available for figure 8:

**Source data 1.** *Figure 8* source data.
DOI: https://doi.org/10.7554/eLife.40470.035

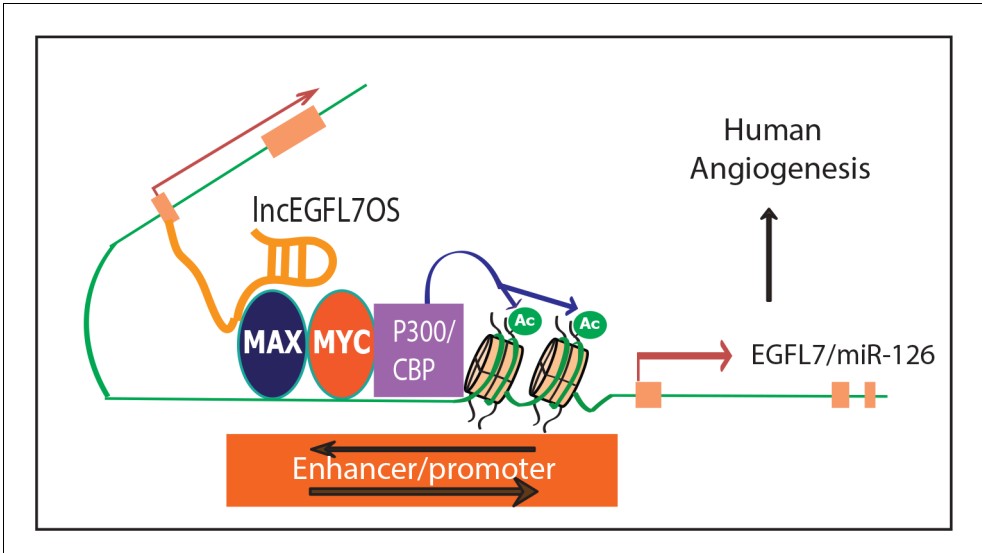

**Figure 9.** A model for lncEGFL7OS in human angiogenesis.  LncEGFL7OS is transcribed in the opposite strand of EGFL7/miR-126 gene under the control of an ETS transcription factors-regulated bidirectional promoter. In turn, lncEGFL7OS transcripts recruit MAX, which interacts with p300 and increase the acetylation of Histone H3K27. This in turn enhances the transcription of EGFL7/miR-126 gene and therefore angiogenesis through MAPK and AKT pathways in human ECs.

DOI: https://doi.org/10.7554/eLife.40470.036

126 in angiogenesis was demonstrated by loss-of-function studies in both mouse and zebrafish. Targeted deletion of miR-126 in mice or miR-126 knockdown in zebrafish resulted in loss of vascular integrity and defective angiogenesis, while overexpression of miR-126 regulates angiogenesis in a cell-type and strand-specific manner (*Fish et al., 2008*; *Wang et al., 2008a*; *Kuhnert et al., 2008*; *Zhou et al., 2016*). It is intriguing that, in contrast to EGFL7 and miR-126, lncEGFL7OS represents a human/primate-specific mechanism in regulating angiogenesis, since lncEGFL7OS only exists in human and several other primates. New angiogenesis mechanism through lncEGF7OS has evolved during evolution, which underscores the importance and delicacy of EFGL7/miR-126 locus in angiogenesis. This study also highlights the importance of using human (and/or primate) system to study the mechanism of angiogenesis.

## Mechanism of lncEGFL7OS action

We showed that the action of lncEGFL7OS reflects at least partially the regulation of expression of EGFL7 and miR-126. miR-126 has been shown to promote MAP kinase and PI3K signaling in response to VEGF and FGF by targeting negative regulators of these signaling pathways, including Spred-1 and PIK3R2. Consistent with the downregulation of miR-126 by lncEGFL7OS silencing, we found that the phosphorylation of ERK1/2 and AKT in response to VEGF is repressed by lncEGFL7OS silencing. Mechanistically, MAX transcription factor was identified as a lncEGFL7OS interaction protein required for lncEGFL7OS-regulated gene expression and angiogenesis in ECs. Under normal conditions, the lncEGFL7OS/MAX interaction is likely locus dependent since several other MAX target genes were not affected by lncEGFL7OS silencing. This is possibly due to the low expression of lncEGFL7OS. LncEGFL7OS enhances the transcription of EGFL7/miR-126 by binding to MAX protein that is recruited to the bidirectional promoter/enhancer region in EGFL7/miR-126. MAX knockdown blunts the induction of miR-126 by lncEGFL7OS in ECs. MAX transcription factor has been shown to interact with MYC to control cell proliferation and cell death (*Amati and Land, 1994*). MYC has been shown to stimulate histone acetylation and gene transcription by recruitment of cAMP-response-element-binding protein (CBP) and p300 (*Vervoorts et al., 2003*). Based on our results showing interaction of MAX and p300, the enrichment of H3K27 acetylation by lncEGFL7OS likely result from the recruitment of CBP and P300 by MAX/MYC. Taken together, lncEGFL7OS acts

in cis by interacting with MAX transcription factor to enhance H3K7 acetylation and promote EGFL7/miR-126 expression.

## Therapeutic implications

Identifying angiogenic mechanisms that are conserved to human is critical for developing therapeutics for human vascular disorders. Our studies have demonstrated that lncEGFL7OS is a human/primate-specific lncRNA critical for human angiogenesis. This may be directly translatable for human diseases involving abnormal angiogenesis. Our studies showed increased expression of both lncEGFL7OS and EGFL7/miR-126 in the heart of DCM patients. Although the causative role of lncEGFL7OS in DCM is still unclear, lncEGFL7OS upregulation may reflect the compensatory vascularization/angiogenesis in DCM. It would be intriguing to test whether manipulating the lncEGFL7OS/EGFL7/miR-126 axis has therapeutic benefits for DCM patients. AMD is the leading cause of blindness in the elderly, and choroidal neovascularization is a hallmark for wet AMD (*Jager et al., 2008*). Although anti-VEGF agents can markedly improve the clinical outcome of wet AMD, they have been unable to induce complete angiogenesis regression, and only 30–40% of individuals experienced vision improvement after treatment (*Folk and Stone, 2010*; *Krüger Falk et al., 2013*). We developed a human choroid sprouting angiogenesis model and showed that silencing of lncEGFL7OS represses human choroid sprouting angiogenesis. It would be appealing to develop and test lncEGFL7OS-based therapy to treat choroidal neovascularization in wet AMD and other vascular disorders in the future. In this regard, our data that CRISPR-mediated targeting of EGLF7/miR-126/lncEGFL7OS locus inhibits angiogenesis could have therapeutic implications in angiogenesis-related diseases. Targeting this locus could be a potent approach for inhibiting angiogenesis than targeting the three genes individually.

## Materials and methods

### Animals and in vivo angiogenesis assay

Animal studies were conducted in accordance with the ARVO statement for the Use of Animals in Ophthalmic and Vision Research and were approved by the Institutional Animal Care and Use Committees at the Tulane University. BALB/cAnN-nu (Nude) female mice (6 to 8 weeks of age) from Jackson lab were used for in vivo angiogenesis assay. In vivo Matrigel analysis was performed as described (*Skovseth et al., 2007*). HUVEC cells transfected with control si-RNA, or mix of si-LncEGFL7OS#1 and si-LncEGFL7OS#2 (50nM each) for 2 days. Cells were then trypsinized and about $5 \times 10^5$ cells were mixed with 50 µl EBM-2 medium and 350 µl ice-cold Matrigel (BD Biosciences). The mixture was then applied under the back skin of 8 week-old BALB/cAnN-nu (Nude) female mice (Jackson lab). After 14 days, The Matrigel plugs were extracted and snap-frozen in OCT and processed for immunostaining with human EC marker PECAM-1 (DAKO), mouse red blood cell marker Ter-119 (Thermo Fisher), mouse smooth muscle marker αSMA (Abcam), and tube length quantification using image J (National Institute of Health).

### Cell culture and siRNAs used

HUVEC (ATCC) cells were grown in EC growth medium EGM-2 (Lonza). HCEC and HREC cells were kindly provided by Dr. Ashwath Jayagapol from Vanderbilt University and grown in EGM2 media (Lonza). EC identity of cells has been confirmed by immunostaining and acetyl-LDL uptake assay (*Figure 1—figure supplement 1*). ARPE-19 (ATCC) cells were growth in DMEM/F12 (HyClone) media with 10% FBS. HDF (ATCC) cells were grown in DMEM (HyClone) with 10% FBS. All cells have been tested negative for mycoplasma contamination. For VEGF treatment, HUVECs were starved with EC basal medium-2 with 0.1% FBS for 24 hr and then treated with VEGF (20 ng/mL) for the indicated periods of time. SiRNA transfection in cell culture was performed as described (*Zhou et al., 2014*). SiRNAs for LncEGFL7OS were purchased from sigma. Sequences for siRNAs are as follows: si-lncEGFL7OS#1: 5'-GCGUUUCCCUAGCAAUGUUdTdT-3' and 5'-AACAUUGCUAGGGAAACGCdTdT-3'; si-lncEGFL7OS#2: 5'-CAGCUUUGCCCUAUCCCAUdTdT-3' and 5'-AUGGGAUAGGGCAAAGCUGdTdT-3'. Two pair of siRNAs for MAX gene include: 5'-CCAGUAUAUGCGAAGGAAAdTdT-3' and 5'-UUUCCUUCGCAUAUACUGGdTdT-3', 5'-CACACACCAGCAAGAUAUUdTdT-3' and 5'-AAUAUCUUGCUGGUGUGUGdTdT-3'. SiRNAs for ETS1 include: 5'-CCGACGAGUGAUGGCAC

UGAAdTdT-3' and 5'-UUCAGUGCCAUCACUCGUCGG-3'. SiRNAs for ETS2 include: 5'-CAGUCA UUCAUCAGCUGGA[dT][dT]−3' and 5'-UCCAGCUGAUGAAUGACUG[dT][dT]−3'.

## LncRNA microarray

RNAs from five cell lines were purified by mirVana$^{Tm}$ total RNA Isolation Kit (Ambion, Invitrogen). These RNAs were subjected to microarray-based global transcriptome analysis (Arraystar Human LncRNA array (version 2.0), Arraystar Inc, Rockville, MD). The lncRNA microarray is designed to detect about 30,586 LncRNAs and 26,109 coding transcripts. The lncRNAs were constructed using the most highly-respected public transcriptome databases (Refseq, UCSC known genes, Gencode, etc), as well as landmark publications. The lncRNA probes include 19590 intergenic lncRNAs (lincR-NAs), 4409 intronic lncRNAs, 1299 bidirectional lncRNAs, 1597 sense overlapping lncRNAs and 3691 antisense lncRNAs. Data analyses, including hierarchy clustering analysis and functional enrichment analysis, were performed using Genescript software. The data have been deposited into NCBI GEO database (GSE105107). Tissue distribution data of the top-50 candidates was downloaded from the Stanford Source database (*Diehn et al., 2003*).

## lncEGFL7OS-expressing adenovirus generation and infection

LncEGFL7OS-, lncEGFL7OS-F(2 + 3), miR-126-, GFP-, or LacZ-expressing adenoviruses were generated as described (*Wang et al., 2008b*). Briefly, lncEGFFL7 cDNA was amplified by PCR using Phusion High-Fidelity DNA Polymerase from HUVEC cDNAs (ThermoFisher Scientific) and cloned into TOPO vector using the following primers: lncEGFL7up: 5'-GCCCTTTGGGCTCAGGCCCAGA-3' and lncEGFL7dn: 5'-GCCCTTTGGGTTTGAGTAATAATTAC-3'. After confirmation by sequencing, the fragment was cloned into pshuttle-CMV vector after HindIII/XhoI digestion. For lncEGFL7OS-F(2 + 3) cloning into pshuttle-CMV vector, the following primers were used: 5'-aaaagatctATGGCGTGTGAG TGCATGGCGAGC-3' and 5'-tataagcttTGGGTTTGAGTAATAATTACATCAT-3'. For making miR-126 adenovirus, miR-126-containing genomic DNA was PCR amplified from mouse using the following primers: 5'-ATGCGAATTC GAGTGAAAGAGCCCCACACTG-3' and 5'-ATGCAAGCTT AG TGCCAGCCGTGGTCCTTAC-3', and cloned into pshuttle-CMV vector after ECORI/HindIII digestion. The positive clones were cut with PmeI and transformed into *E. coli* with adenovirus vector for recombination. Positive clones were then cut with PacI and transfected into Ad-293 cells using Viral-Pack Transfection Kit from Stratagene. Viral titers were determined by End-Point Dilution Assay. For adenovirus infection, the cells were switched to serum free EBM-2 medium and adenovirus was added at an MOI of 10. The infection medium was removed after 3 hr. Cells were washed with PBS and overlaid with fresh growth medium and cultured for 48 hr before further experiments.

## Cell proliferation, Cell cycle analysis, TUNEL assay, Scratch-Wound, and in vitro EC-fibroblast co-culture angiogenesis assay

EC cell proliferation, TUNEL assay and scratch-wound assays were performed using HUVEC cells as described (*Zhou et al., 2014*). For cell proliferation assay, about $2 \times 10^3$ transfected HUVECs were seeded in 96-well plates. After starvation with 0.1% serum for overnight, the cells were stimulated with 20 ng/mL VEGF-A for 20 hr and then subjected to BrDU labeling for 4 hr. DNA synthesis as determined by BrDU incorporation was quantified using a commercial ELISA kit from Roche according to the manufacturer's instructions. Cell cycle analysis was performed using Guava Cell Cycle Reagents (Guava Technologies) on a Guava instrument and analyzed using Cytosoft software according to the manufacturer's manual. For scratch wound assay, scratch-wound was made using a 200 μL pipette tip in lncRNA or control siRNA–transfected HUVEC monolayer before VEGF (20 ng/mL) stimulation. 1 μM of 5-fluouracil (Sigma) was then added to the cells right after scratch wound to block cell proliferation. Post-scratch EC migration was scored at 14 hr after wound scratch. For in vitro angiogenesis assay, at 3 days after lncRNA or control siRNA transfection with Liptofectamine RNAi-MAX reagent (Invitrogen), cells were harvested for RNA or in vitro Matrigel assay and branch point analysis as described before.

In vitro EC-Fibroblast co-culture was performed as described (*Hetheridge et al., 2011*). Briefly, human dermal fibroblast cells (HDF) were seeded into each well of a 24 well plate and maintained in DMEM at $6 \times 10^3$ cells/well until they developed confluent monolayers. HUVECs were maintained as described above and transfected with siRNA one day prior to seeding on HDF monolayers.

Approximately $6 \times 10^3$ HUVECs were seeded onto each monolayer and the HDF/HUVEC co-culture was maintained for 7 days in EGM-2 medium with medium changes every 2–3 days to allow endothelial cell polarization, migration, networking, and the formation of an in vitro primitive vascular plexus. For rescue experiments, some wells were transfected with Ad-miR-126 (MOI of 10) and EGFL7 (Abcam) protein was added to the medium at 10 nM every other day. After 7 days the wells were fixed with 100% Methanol at $-20°$ C for 20 min and then stained with anti-PECAM-1 (DAKO). After hybridizing a secondary antibody, the endothelial tissue was visualized and imaged under a Nikon microscope. Multiple images were automatically stitched with Nikon software to provide a large image (several mm [*Roura et al., 2007*]) and the resulting image was analyzed on ImageJ software to determine the degree of vascularization. Three wells were used for each condition and results are representative of the mean of each three well group. The experiments were repeated for at least three repeats with similar results.

## Ex vivo human choroid sprouting assay

Ex vivo human choroid sprouting assay was adapted from a mouse protocol (*Shao et al., 2013*). Donated human eye balls were obtained from Southern eye bank (New Orleans, LA). The use of deceased human eye balls for the study was EXEMPT under DHHS regulations (46.101(b)) after consultation with the Tulane IRB committee. Informed consent has been obtained from all subjects by Southern eye bank. Eyes were collected within 24 hr of decease of the donors, and cleaned and kept in sterile ice-cold PBS with Penicillin/Streptomycin before dissection. Using fine forceps, the cornea and the lens from the anterior of the eye were removed. The peripheral choroid-scleral complex was separated from the retina and the RPE layer was peeled away using fine forceps. The choroid-scleral complex was then cut into approximately 4 mm$^2$ pieces using sterile scalpel blade under laminar airflow. The choroid was then washed with sterile ice-cold PBS and transferred into endothelial base medium (EBM2) with 0.1% FBS (300 µl/well in 24-well plates). The choroid was transfected with control si-RNA, or mix of si-LncEGFL7OS#1 and si-LncEGFL7OS#2 (50nM each) for overnight. Choroid fragments were then washed by EGM2 media then placed in growth factor-reduced Matrigel$^{TM}$ (BD Biosciences) in 24-well plate. Briefly, 30 µl of matrigel was used to coat the bottom of 24 well plates without touching the edge of the well. After seeding the choroid, the plate was incubated in a 37 °C cell culture incubator to make the Matrigel solidify. 500 µl EC growth medium (EGM-2) were added slowly to the plate without disturbing the Matrigel, and the plate was incubated at 37 °C cell culture incubator with 5% $CO_2$. Cell culture medium was changed every 48 hr. The EC sprouts normally start to appear on the day five and grow rapidly between day 7 and 10. Phase contrast photos of individual explants were taken using a Nikon microscope. The sprouting distance was quantified with computer software ImageJ (National Institute of Health). Sprouting ECs were stained with ICAM-2 (BD Pharmingen) or isolectin B4 (Vector Lab).

## RNA, Western blot analysis and Immunofluorescence

Human total RNA master panel II was purchased from clontech (Takara). Total RNA was isolated from human choroid tissues or cell lines using TRIzol reagent (Invitrogen). Cytoplasmic and nuclear RNA was purified using a Cytoplasmic and Nuclear RNA Purification Kit (Norgen Biotek Corp., Thorold, ON, Canada) according to manufacturer's supplied protocol. In brief, cells growing in monolayer were rinsed with 1xPBS and lysed directly on the plate with ice-cold Lysis Buffer. Next cell lysate was transferred to the RNase-free microcentrifuge tube and spun for 3 min at 14,000 x g. Supernatant containing cytoplasmic RNA was mixed with manufacturer's supplied buffer (Buffer SK) and 100% ethanol, and applied onto a spin column. The pellet containing the nuclear RNA was mixed with Buffer SK and 100% ethanol, and applied onto a spin column. Both columns were washed with supplied Wash Solution, and RNA was eluted with supplied elution buffer (Elution Buffer E). For maximum recovery two rounds of elution were performed. Quantitative (q) RT-PCR or regular RT-PCR was performed using iScript cDNA Synthesis system (BioRad), miRNA qRT-PCR was performed using qScript cDNA Synthesis and microRNA Quantification System (Quanta Biosciences). lncEGFL7OS RACE PCR was performed using Marathon –ready cDNA from human placenta (Clontech, Mountain View, CA). 5'RACE and 3'RACE PCR was carried out using lncEGFL7OS primers and primers from the kit. Then a second round of PCR was performed using the combination of the RACE products and the RACE primers from the kit. The derived PCR product was then cloned using

TOPO vector and sequenced. Primers for real-time PCRs include human β-actin, 5'-GAGCAAGAGA TGGCCACGG-3' and 5'-ACTCCATGCCCAGGAAGGAA-3'; lnc-FLI1-AS1 (also named SENCR), up: 5'- CCTGAGGCCATCTTACCACC-3', down: 5'- AATCCGCTTCGATGAGTGGG-3'; SENCR (for regular PCR), up: 5'-GCGCATTGTTAGGAGAAGGG-3', down: 5'- CCTGCTGACTGTCCTAGAGG-3'; lnc-GATA2-AS, up: 5'-CGGGCAGCTTACGATTCTTC-3', down: 5'- CGGTGTCTTTCAGAGGGTCT-3'; lnc-ECE1, up: 5'- CCATGTCGCCTCAGCCTAAA−3', down: 5'- GGGCAGTCTCAGGGTAACAC-3'; lnc-ESAM, up: 5'-CTCGGAAAACGGAGGGTTGA-3', down: 5'- CGCTGCCCTTAATTCCTTGC-3'; lnc-ROBO4-AS, up: 5'- ACCAGCAGACCCTGAAACTC-3', down: 5'-GGCAGGGATCAGGCATTCAT-3'; lnc-EGFL7OS, up: 5'- AGTGCCAGCTTTGCCCTATC-3', down: 5'- GAGAACACAGGACGTCCACA-3'; EGFL7-A, up: 5- CTTCAGAGGCCAAAAGCACC-3', down: 5'- GAATCAGTCATCCCCCGGAC-3'; EGFL7-B, up: 5'- AAGGGAGGCTCCTGTGGA-3', down: 5'- CCTGGGGGCTGCTGATG-3'; EGFL7-C, up: 5'- CGGATCCGGCGGCCA-3', down: 5'- CGAACGACTCGGAGACAGG-3'; Neat1, up: 5'-AGA TACAGTGTGGGTGGTGG-3', down: 5'-AGTCTTCCCCACCTTGTAGC-3'. Human primiR-126, up: 5'-TGGCGTCTTCCAGAATGC-3', down: 5'-TCAGCCAAGGCAGAAGT-3'. Human Cyclin D2, up: 5'-GC TGTGCATTTACACCGACA-3'; down: 5'-TGCGCAAGATGTGCTCAATG-3'. Human DFHR, up: 5'-A TTTCGCGCCAAACTTGACC-3'; down: 5'-TCTGAATTCATTCCTGAGCGGT-3'.

For western blot analysis, protein lysates were resolved by SDS-PAGE and blotted using standard procedures. Antibodies used were as follows: ERK1/2 (Cell signaling), Phospho-ERK1/2 (Cell signaling), AKT (Cell signaling), Phospho-AKT (Cell signaling), EGFL-7(Abcam) and β-Tublin (Abcam). For immunofluorescence experiments, samples were fixed with 4% paraformaldehyde or methanol for 30 min. After treatment with 1% Triton X-100 in PBS, samples were incubated in PBS containing 4% goat serum for 30 min. The samples were then incubated with primary antibodies overnight at 4°C, followed by incubation with appropriate secondary antibodies. Antibody used for immunofluorescence include: ICAM-2(BD Pharmingen), PECAM-1 (DAKO).

## Single-cell RNA copy number determination

Single-cell lncEGFL7OS RNA copy number was determined as modified from a previous publication (*Wagatsuma et al., 2005*). Briefly, $10^6$ HUVEC cells were harvested for total RNA isolation using Trizol. 16% (8µl out of 50 µl) of the total RNA was used for reverse transcription reaction as described above, and 1/100 of the cDNA was used as template in each well for the subsequent qRT-PCR. Therefore, for each well, the total lncEGFL7OS came from about ~1600 cells. For establishing the standard curve, pCRII-TOPO-lncEGFL7OS plasmid was linearized for generating lncEGFL7OS RNA by in vitro transcription. After concentration determination and copy number calculation, a given amount of RNA was employed to carry out the reverse transcription under the same conditions for HUVEC total RNA. The derived cDNA was diluted for PCR to generate a standard curve for lncEGFL7OS PCR. The copy number of RNA per cell was calculated based on the CT number (*Supplementary file 4*).

## High resolution RNA FISH experiments

25 Stellaris RNA Fluorescence In Situ Hybridization (FISH) probes for lncEGFL7OS were designed according to Stellaris FISH probe designer (https://www.biosearchtech.com/ Account/Login? return=/stellaris-designer) (BiosearchTech, *Supplementary file 5*). RNA-FISH was performed following the manufacturer's protocol. Briefly, HUVECs cultured on 18 mm coverglasses were fixed and permeabilized by methanol-acetic acid solution for 10 min. After removing the fixation solution, cells were washed by Wash Buffer A (Biosearch Tech) at room temperature for three minutes, and then transferred to a humidified chamber to incubate with Hybridization Buffer (Biosearch Tech) containing the probes. The coverglasses were put upside-down on Parafilm for overnight. After washing with Wash Buffer A (Biosearch Tech) at 37°C for 30 min, the cells were incubated with Wash Buffer A containing 5 ng/ml DAPI in the dark at 37°C for 30 min. Finally, Wash Buffer B was added and the cells were incubated at room temperature for 5 min before mounting coverglass onto the slides with mounting medium. Pictures were taken under a Nikon A1 confocal microscope. For RNA copy quantification, hybridization signals and DAPI positive nucleus were counted manually.

## Co-immunoprecipitation (Co-IP), Chromatin Immunoprecipitation (ChIP) and RNA immunoprecipitation (RIP) assays

Co-immunoprecipitation assay was carried out following the Abcam protocol. Briefly, $10^7$ HUVEC cells were scraped and resuspended in ice-cold lysis buffer (20 mM Tris.Hcl pH8, 137 mM NaCl, 1% NP-40, 2 mM EDTA,10mM beta-mercaptoethanol, 15 U/ml DNAse I, protease Inhibitors). After 30 min on ice, cell lysate was centrifuged at 12000 g for 15 min at 4°C. The supernatant was transferred to another pre-chilled tubes and pre-cleared by 2 μg off-target rabbit antibody (Santa Cruz) followed by 40 μl of protein G magnetic bead slurry (Bio-rad) at 4°C. 25 μl pre-cleared cell lysate was reserved as input control. The rest was divided into two parts and added 2 μg of off-target rabbit IgG (Santa Cruz) and anti-P300 antibody (Abcam) respectively. The samples were incubated with antibodies at 4°C for overnight under gentle rotation. Then, 60 μl of protein G magnetic bead slurry (Bio-rad) was added into each sample. Incubate the lysate beads mixture at 4°C under rotation for 4 hr, then centrifuge the tubes and discard supernatant. The beads were washed with lysis buffer gently for three times. The proteins were eluted by SDS loading buffer (supplemented with 10 mM beta-mercaptoethanol and protease Inhibitors). Western blot was used to analyze the content of samples.

ChIP experiments were performed as described with some modifications (*Nelson et al., 2006*). Briefly, HUVEC cells were cultured in the 10 cm dishes to 80–90% of confluence. After adding 400 μl of 37% formaldehyde to 10 ml medium and incubation for 15 min to fix the cells, cells were rinsed by pre-chilled PBS buffer and collected in 1 ml IP buffer (150 mM NaCl, 50 mM Tris-HCl (pH 7.5), 5 mM EDTA, NP-40 (0.5% vol/vol), Triton X-100 (1.0% vol/vol),1% proteinase inhibitor cocktails). After half an hour of sonication, 2 μg of antibodies were added into cell lysate and incubated in ultrasonic bath for 30 min. Protein G Magnetic Beads were used to pull down antibodies in 4°C rotating platform for 2 hr. Once beads were washed for 5 times by cold IP buffer, 100 μl 10% (wt/vol) Chelex-100 was mixed with washed beads, and the mixture was boiled for 10 min. Each sample was added 1 μl of 20 μg/μl proteinase K and incubated at 55°C for 30 min. Samples were boiled for 10 min again and centrifuged. Supernatant were collected for real-time PCR. ChIP grade antibodys used in ChIP assay: Max (Santa Cruz, sc-197), Myc (Sigma-Aldrih, c3956), Anti-RNA Polymerase II (Abcam, ab5408), Tri-Methel-Histon H3(Lys4) (Cell Signaling, #9751), ETS1(Santa Cruz, sc-111), H3K27Ac antibody (Abcam, Ab4729), Normal Rabbit IgG (Cell Signaling, #2729). ChIP samples were analyzed by using normal PCR with following parameters: (1) initial denaturation at 94°C for 10 min, (2) denaturation at 94°C for 20 s, (3) anneal at 58°C for 30 s, (3) extension at 72°C for 1 min. Steps from 2 to 4 were repeated 35 times. Primers to amplify conserved transcription factors binding region in the lncEGFL7OS enhancer/promoter region were as follows: Primers 1: 5′- CTGGCTGTTTTGGGGC TAGA-3′ and 5′- CCTGTGTGTGTTCTCCGCT-3′. Primers 2 (control region): 5′- AGATCCCAGGGC TGTTTAGC-3′ and 5′- AACACTCCTCCCAGCGAATC-3. Primers for Cyclin D2 and DFHR promoter regions are as follows: Cyclin D2 promoter-F: 5′-GCAGGGAACCTAGTGTACGG-3′; Cyclin D2 promoter-R: 5′-CGCGCCCTTTGGTGTATTTC-3′; DHFR promoter-F: 5′-CGGGGCTACAAATTGGGTGA-3′; DHFR promoter-R: 5′-TAAAAGACGCACCCCTTGCC-3′.

RNA immunoprecipitation (RIP) was performed following a protocol from Abcam. Briefly, $10^7$ Ad-GFP or Ad-lncEGFL7OS-infected HUVEC cells were harvested by trypsinization, and resuspended in PBS buffer respectively when the confluence was about 90%. Freshly prepared nuclear isolation buffer (1.28 M sucrose, 40 mM Tris-HCL pH7.5, 20 mM $MgCl_2$, 4% Triton X-100) was diluted by 3× ddH$_2$O and used to resuspend the above cell pellets. After incubation on ice for 20 min with frequent mixing, cell nuclei were collected by centrifugation at 2500 g for 15 min at 4°C, and resuspended in 1 ml freshly prepared RIP buffer (150 mM KCl, 25 mM Tris pH7.4, 5 mM EDTA, 0.5 mM DTT, 0.5% NP40, 100 U/ml RNAase inhibitor, protease inhibitors). After chromatin shearing, RNA supernatants were collected by centrifugation at 13000 rpm for 10 min to remove nuclear membrane and debris. 2 μg mock and anti-Max IgG were added into 500 μl supernatant respectively and incubated overnight at 4°C. 40 μl protein G magnetic beads (Bio-rad) was added and incubated for 2 hr at 4°C with gentle rotation. Coprecipitated RNAs were resuspended in 1 ml TRIzol reagent (Invitrogen) according to manual. Extracted RNAs were employed for subsequent reverse transcription and cDNA analysis. Some RNA samples were used as controls.

## Determination of MAX binding domain in lncEGFL7OS

LncEGFL7OS was separated into three domains according to its predicted secondary structure. Briefly, F1 domain contains 1-239nt of lncEGFL7OS, F2 domain contains 208-393nt and F3 domain contains 377-557nt. The separated domains were PCR amplified and sub-cloned into pShuttle-CMV vectors (Agilent Technologies) respectively, and transfected into APRE-19 cells together at 3 μg per vector per dish. After 48 hr, cells were harvested, the expression of the lncRNA fragments was confirmed by RT-PCR, and RNA immunoprecipitation(RIP) was performed by using MAX antibody as described above. Wild type ARPE 19 cells were harvested as background control since its lncEGFL7OS level is under the detection threshold. Dnase I was used to remove potential DNA contamination from the RNA samples before first-strand cDNAs were synthesized. Primers for construction and detection as below: F1-5': 5'-AATAGATCT TGGGCTCAGGCCCAGAGTGCCA-3'; F1-3':5'-AAAAAGCTT CT GGAGGCGCTCGCCATGCAC-3'; F2-5': 5' AATAGATCT ATGGCGTGTGAGTG CATGGC-3'; F2-3': 5'-AAAAAGCTT TCAGGTAGCTGCGAGTTCAAG-3'; F3-5': 5'-AATAGATCTAC TCGCAGCTACCTGAGTCAGA-3'; F3-3': 5'-AAAAAGCTT TG GGTTTGAGTAATAATTACATC-3'.

## CRISPRi (dCas9-KRAB) Assay

CRISPRi (dCas9-KRAB) assay was perform as described (*Larson et al., 2013*). pHR-SFFV-dCas9-BFP-KRAB (Addgene:46911) and control (pLJM1-EGFP) vectors were packaged into lentivirus vectors respectively. sgRNA-1(TGCTTACAGGCAAGGGGCGA) and sgRNA-2 (AAGAATTGCTTCAGC TCGGA), which target lncEGFL7OS promoter and intron respectively, were subcloned into lenti-Guide-Puro vector (Addgene: 52963), which could express sgRNAs to assemble with dCas9-Krab. Empty lentiGuide-Puro vector serves as control. For the assay, HUVEC cells were transduced by control or dCas9-Krab vector, combing with lentiGuide-gRNA1, lentiGuide-gRNA2 and empty lenti-Guide-Puro, respectively. All lentivirus vectors were employed at 10 MOI. EC-fibroblast co-culture was performed as described above.

## Luciferase assay

Luciferase assays were performed as described (*Wang et al., 2008a*). The putative bidirectional promoter for *lncEGFL7OS/EGFL7* was PCR amplified from human DNA and cloned into promoterless PGL3 Basic luciferase vector (Promega). Primers include: plncEGFL7OSup (XhoI): 5'-atcgCTCAGA TAGACTCTGATGGCCCAGG-3' and plncEGFL7OSdn (XhoI): 5' –atcgCTCAGACCAGCTTGG TGCAGGGAG-3'. 293 T cells in 24-well plates were transfected with 50 ng of reporter plasmids in the presence or absence of increasing amount of Ets1 or Ets1 DNA-binding mutant expression plasmid.

## Human samples

The human study was performed according to the principles of the Declaration of Helsinki. Patient information was described previously (*Huang et al., 2015*). The procedure was approved by the Institutional Ethics Committee of the National Institute of Cardiovascular Diseases, Bratislava, Slovakia. Briefly, left ventricular tissues from seven patients with terminal-stage heart failure and five control healthy donors were dissected and snap frozen, and used for RNA isolation and gene expression study.

## Statistics

In the bar graphs without P-value analysis, the central values are the means, and the error bars are standard deviation. In the bar graphs with P-value analysis, the central values are the means, and the error bars are standard error of means. Significant differences between groups were analyzed via Student's unpaired t-test (default). For multiple group analysis, significances between multiple groups were analyzed by ordinary ANOVA followed by Tukey honest significant difference testing. P-values of less than 0.05 were considered to be statistically significant.

## Acknowledgements

We thank Ms. Joy Roussel, Mr. Adam Leimer and other members from Southern Eye Bank for their generous support. We thank Dr. Ming Zhan from Houston Methodist Research Institute for help and

discussion of the microarray data, Dr. Joseph Miano from University of Rochester Medical Center for help in RNA FISH experiments, and Dr. Frank Jones from Tulane University for sharing equipment. SW was supported by a startup fund from Tulane University, NIH Grants EY021862 and EY026069, a career development award from the Research to Prevent Blindness foundation, and a Bright Focus Foundation Award in Age-related Macular Degeneration. QZ was supported by an American Heart Association (AHA) postdoctoral fellowship. KZ was supported by grant 2G12MD007595-07 from National Institute on Minority Health and Health Disparities (NIMHD), NIH and Department of Health and Human Services (DHHS). P B was supported by 2G12MD007595-06 from NIMHD, NIH diversity consortium program/Building Infrastructure Leading to Diversity (BUILD)−2 T34GM007716-38.

## Additional information

### Funding

| Funder | Grant reference number | Author |
|---|---|---|
| American Heart Association | Postdoctoral Fellowship | Qinbo Zhou |
| National Institute on Minority Health and Health Disparities | 2G12MD007595-06 | Partha S Bhattacharjee |
| National Institutes of Health | (BUILD)-2T34GM007716-38 | Partha S Bhattacharjee |
| National Institutes of Health | 2G12MD007595-07 | Kun Zhang |
| National Institutes of Health | | Kun Zhang |
| U.S. Department of Health and Human Services | | Kun Zhang |
| National Eye Institute | EY021862 | Shusheng Wang |
| National Eye Institute | EY026069 | Shusheng Wang |
| Research to Prevent Blindness | Career Development Award | Shusheng Wang |
| BrightFocus Foundation | Award in Age-related Macular Degeneration | Shusheng Wang |

The funders had no role in study design, data collection and interpretation, or the decision to submit the work for publication.

### Author contributions

Qinbo Zhou, Conceptualization, Data curation, Formal analysis, Investigation, Visualization, Methodology; Bo Yu, Data curation, Investigation, Visualization, Methodology; Chastain Anderson, Jakub Hanus, Wensheng Zhang, Yu Han, Partha S Bhattacharjee, Sathish Srinivasan, Kun Zhang, Da-zhi Wang, Investigation; Zhan-Peng Huang, Data curation, Investigation; Shusheng Wang, Conceptualization, Supervision, Funding acquisition, Investigation, Writing—original draft, Project administration, Writing—review and editing

### Author ORCIDs

Shusheng Wang (iD) http://orcid.org/0000-0002-8841-5432

### Ethics

Human subjects: The human study was performed according to the principles of the Declaration of Helsinki. Patient information was described previously. The procedure was approved by the Institutional Ethics Committee of the National Institute of Cardiovascular Diseases, Bratislava, Slovakia.
Animal experimentation: Animal studies were conducted in accordance with the ARVO statement for the Use of Animals in Ophthalmic and Vision Research and were approved by the Institutional Animal Care and Use Committees at the Tulane University.

Decision letter and Author response
Decision letter https://doi.org/10.7554/eLife.40470.046
Author response https://doi.org/10.7554/eLife.40470.047

## Additional files

### Supplementary files

• Supplementary file 1. List of top-50 EC-enriched lncRNAs and their associated genes.
DOI: https://doi.org/10.7554/eLife.40470.037

• Supplementary file 2. List of EC-enriched enhancer-like lncRNAs from the array.
DOI: https://doi.org/10.7554/eLife.40470.038

• Supplementary file 3. List of the EC-enriched lncRNAs that have associated protein-coding genes within 10 kb, showing parallel or inverse expression pattern with their associated genes.
DOI: https://doi.org/10.7554/eLife.40470.039

• Supplementary file 4. (A) CT values from the PCR using standard in vitro transcribed lncEGFL7OS RNA. The RNA was harvested at $1.85*10^{11}$ copies per µl. After reverse transcription, 1 µl the cDNA was diluted at $10^3$, $10^4$, $10^5$, $10^6$ and $10^7$ times, respectively, as templates to carry out Real-time PCR. The copy numbers were calculated based on the dilution folds. (B) The CT values and the $\log_{10}$ (Copy number) were used to establish the standard curve and formulation for copy number calculation. The $\log_{10}$ (copy number) and CT value relation can be modeled as: $Y = -0.4438*X + 16.15$. R square is 0.9415. (C) The formulation in (B) was used to calculate the copy number per well of the HUVEC cell samples. Based on the calculation that each well has ~1600 cells, the copy number per cell was calculated.
DOI: https://doi.org/10.7554/eLife.40470.040

• Supplementary file 5. LncEGFL7OS Stellaris FISH probes designed according to Stellaris FISH probe designer.
DOI: https://doi.org/10.7554/eLife.40470.041

• Transparent reporting form
DOI: https://doi.org/10.7554/eLife.40470.042

### Data availability

lncRNA microarray data has been uploaded to the GEO database under accession number GSE105107.

The following dataset was generated:

| Author(s) | Year | Dataset title | Dataset URL | Database and Identifier |
|---|---|---|---|---|
| Wang S, Zhou Q | 2018 | Comparative study of mRNAs and lncRNAs in endothelial and non-endothelial cells | https://www.ncbi.nlm.nih.gov/geo/query/acc.cgi?acc=GSE105107 | NCBI Gene Expression Omnibus, GSE105107 |

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
