## [Decision Letter]

Thank you for submitting your article "LncEGFL7OS regulates human angiogenesis by interacting with MAX at the EGFL7/miR-126 locus" for consideration by *eLife*. Your article has been reviewed by three peer reviewers, including Jason Fish as the Reviewing Editor and Reviewer #1, and the evaluation has been overseen by Didier Stainier as the Senior Editor.

The reviewers have discussed the reviews with one another and the Reviewing Editor has drafted this decision to help you prepare a revised submission.

Summary:

The reviewers found your work to be an in-depth mechanistic study that is novel and of significant interest. This manuscript contributes to a growing body of work describing lncRNA function in endothelial cells. The data provide a greater understanding of the EGFL7/miR-126 locus, and is an important contribution to the field of vascular biology. Furthermore, these data contributes to the understanding of lncRNA mechanisms, especially regarding cis and in trans regulation. The reviewers commented that overall the experiments were well controlled and that the data support the main conclusions. However, there are several additional experiments that would provide necessary support for your conclusions.

Essential revisions:

1) The conclusion regarding the ability of lncEGFL7OS to recruit MAX to the EGFL7 locus is made from lncEGFL7OS over-expression studies. These ChIP experiments (MAX, H3K27ac) should be performed in cells in which lncEGFL7OS is knocked-down to ensure that this happens endogenously. LncEGFL7OS over-expression appears to increase MAX recruitment to multiple MAX target sites that are outside of the EGFL7 locus. It is important to demonstrate the specificity by also assessing non-MAX target sites.

2) For the in vivo Matrigel implantation model depicted in Figure 3C, the authors show increased PECAM-1 labelling indicating endothelial networking activity. A major foundation of this manuscript is that lncEGFL7OS regulates human angiogenesis. While many other assays show angiogenic behaviour, the assay in Figure 3C could show actual blood vessel formation and blood flow and would strengthen the findings. For example, a stain for a red blood cell marker or contrast within the Matrigel implant could definitively show "angiogenesis".

3) Figures 2C and 2D show the subcellular localization of the lncEGFL7OS RNA. These results provided a basis to study the nuclear function of lncEGFL7OS. However, for Figure 2D, only one cell is shown in each image. In each of these images, two fluorescent points are seen in the nucleus. It is possible that these indicate FISH probes hybridize to genomic DNA and the Materials and methods section does not indicate any RNAse A treatments as a negative control or DNAse treatments as a positive control. Having images with these controls or showing cells with more nuclear signal could improve the strength of conclusions (e.g. Cabili et al., 2015).

4) The authors determine that the 5' portion of the lncRNA binds to MAX, and that lncEGFL7OS RNA overexpression increases MAX binding. However, the authors do not show directly that binding of lncEGFL7OS RNA to MAX is required for increased expression of EGFL7 mRNA/miR-126 and whether the binding stimulates angiogenesis. This is a crucial experiment missing in the manuscript and will be an important test for the model presented in Figure 8. This could be achieved by over-expressing a lncEGFL7OS transcript that has been engineered to not bind MAX.

5) It is unclear how low the expression levels of the lncEGFL7OS really are in endothelial cells when compared to other lncRNAs, Egfl7 mRNA and miR-126. Very low expression in HUVEC by RNA FISH is shown in Figure 2D. This is an important issue to assess its in vivo relevance and should be determined.

[Editors' note: further revisions were requested prior to acceptance, as described below.]

Thank you for resubmitting your work entitled "LncEGFL7OS regulates human angiogenesis by interacting with MAX at the EGFL7/miR-126 locus" for further consideration at *eLife*. Your revised article has been favorably evaluated by Didier Stainier as the Senior Editor, and a Reviewing Editor.

The manuscript has been improved but there are some remaining issues that need to be addressed before acceptance, as outlined below:

1) The authors refer to copy number determinations for the lncRNA using a standard curve, but these data are not included in the manuscript. The data should be added, possibly as a supplement to the relevant figure.

2) In the figures that contain screen shots of the UCSC Genome Browser, it is still not clear what cell types the different colors of tracks indicate. This information should be added to the figure legend text.

3) In Figure 5B, the authors include miR-126 data that is normalized to U6 and miR-24. It would be preferable to include miR-126 data normalized to U6 and miR-24 data normalized to U6 to support the authors' claim that miR-24 levels did not change.

---

## [Author Response]

Essential revisions:1) The conclusion regarding the ability of lncEGFL7OS to recruit MAX to the EGFL7 locus is made from lncEGFL7OS over-expression studies. These ChIP experiments (MAX, H3K27ac) should be performed in cells in which lncEGFL7OS is knocked-down to ensure that this happens endogenously. LncEGFL7OS over-expression appears to increase MAX recruitment to multiple MAX target sites that are outside of the EGFL7 locus. It is important to demonstrate the specificity by also assessing non-MAX target sites.

We have performed MAX, H3K27ac CHIP experiments in lncEGFL7OS knockdown cells, and found that lncEGFL7OS knockdown decreased MAX and H3K27ac recruiting to the EGFL7 locus (as shown by new Figure 7A and 7B). We also include a negative control region for Chip PCR experiments showing the specificity of MAX to the MAX targeting sites (Figure 6—figure supplement 1A).

2) For the in vivo Matrigel implantation model depicted in Figure 3C, the authors show increased PECAM-1 labelling indicating endothelial networking activity. A major foundation of this manuscript is that lncEGFL7OS regulates human angiogenesis. While many other assays show angiogenic behaviour, the assay in Figure 3C could show actual blood vessel formation and blood flow and would strengthen the findings. For example, a stain for a red blood cell marker or contrast within the Matrigel implant could definitively show "angiogenesis".

We have performed a 2-week in vivo Matrigel with the stainings suggested by the reviewers. The results show partial overlapping of human PECAM-1 staining and mouse Ter-119 (red blood cell marker) and α-SMA (smooth muscle marker) staining (New Figure 3C, and Figure 3—figure supplement 2) in the blood vessels formed from the HUVEC cells in vivo. These suggest functionality of the formed vessels.

3) Figures 2C and 2D show the subcellular localization of the lncEGFL7OS RNA. These results provided a basis to study the nuclear function of lncEGFL7OS. However, for Figure 2D, only one cell is shown in each image. In each of these images, two fluorescent points are seen in the nucleus. It is possible that these indicate FISH probes hybridize to genomic DNA and the Materials and methods section does not indicate any RNAse A treatments as a negative control or DNAse treatments as a positive control. Having images with these controls or showing cells with more nuclear signal could improve the strength of conclusions (e.g. Cabili et al., 2015).

We have redesigned new in situ experiments using Stellaris FISH probes (a mix of 25 probes, Biosearch Technologies). We have included RNaseA-treated sample as negative control and Ad-lncEGFL7OS-overexpressed HUVECs as positive control (new Figure 2C). The signals have been quantified The results showed that average HUVEC cells have about 19 copies of lncEGFL7OS RNAs. This results are in agreement with our new real-time PCR experiments showing about 23-28 copies of lncEGFL7OS in HUVEC cells. This information is included in the revised text.

4) The authors determine that the 5' portion of the lncRNA binds to MAX, and that lncEGFL7OS RNA overexpression increases MAX binding. However, the authors do not show directly that binding of lncEGFL7OS RNA to MAX is required for increased expression of EGFL7 mRNA/miR-126 and whether the binding stimulates angiogenesis. This is a crucial experiment missing in the manuscript and will be an important test for the model presented in Figure 8. This could be achieved by over-expressing a lncEGFL7OS transcript that has been engineered to not bind MAX.

Thanks to the reviewer for the great suggestion. We have taken the advice and have done experiments to overexpress lncEGFL7OS-(F2+F3) portion of the RNA, which doesn’t interact with MAX. We found that overexpression of this form doesn’t impact the expression of EGFL7 and miR-126, nor affect angiogenesis in EC-fibroblast co-culture assay. These confirm that the interaction of lncEGFL7OS with MAX is required for EGFL7/miR-126 regulation and angiogenesis enhanced by EGFL7/miR-126.

5) It is unclear how low the expression levels of the lncEGFL7OS really are in endothelial cells when compared to other lncRNAs, Egfl7 mRNA and miR-126. Very low expression in HUVEC by RNA FISH is shown in Figure 2D. This is an important issue to assess its in vivo relevance and should be determined.

Thanks for the important issue raised. To confirm the expression level of lncEGFL7OS in HUVEC cells, we have redesigned the in situ experiment using new Stellaris FISH probes (a mix of 25 probes, Biosearch Technologies, see Supplementary file 5). We have included RNaseA-treated sample as negative control and Ad-lncEGFL7OS-overexpressed HUVECs as positive control (new Figure 2C). The signals have been quantified. The results showed that average HUVEC cells have about 19 copies of lncEGFL7OS RNAs. These results are in agreement with our new qPCR experiments showing about 23-28 copies of lncEGFL7OS in HUVEC cells. This information is included in the revised text.

[Editors' note: further revisions were requested prior to acceptance, as described below.]

The manuscript has been improved but there are some remaining issues that need to be addressed before acceptance, as outlined below:1) The authors refer to copy number determinations for the lncRNA using a standard curve, but these data are not included in the manuscript. The data should be added, possibly as a supplement to the relevant figure.

Standard curve and detailed method for quantification have been added as Supplementary file 4.

2) In the figures that contain screen shots of the UCSC Genome Browser, it is still not clear what cell types the different colors of tracks indicate. This information should be added to the figure legend text.

Eight cell types were tracked in the image. Light blue indicates HUVEC cells, while dark color indicates H7-ES cells.

3) In Figure 5B, the authors include miR-126 data that is normalized to U6 and miR-24. It would be preferable to include miR-126 data normalized to U6 and miR-24 data normalized to U6 to support the authors' claim that miR-24 levels did not change.

The miR-24 data could not be found in our data collections, even though we are confident we have the data somewhere. The author who has been involved in the experiment has left the lab, and he could not find that piece of data either. I feel this is not a critical piece of the data, and has removed the statement from the paper. If the reviewer feels strongly we need to have the data, we can definitely do some more experiments on it.